# EFFICIENT CROSS-EPISODE META-RL

**Gresa Shala[1], André Biedenkapp[1], Pierre Krack[2], Florian Walter[2] & Josif Grabocka[2]**
[1]University of Freiburg, Germany
[2]University of Technology Nuremberg
`shalag@cs.uni-freiburg.de`

## ABSTRACT

We introduce Efficient Cross-Episodic Transformers (ECET), a new algorithm for online Meta-Reinforcement Learning that addresses the challenge of enabling reinforcement learning agents to perform effectively in previously unseen tasks. We demonstrate how past episodes serve as a rich source of in-context information, which our model effectively distills and applies to new contexts. Our learned algorithm is capable of outperforming the previous state-of-the-art and provides more efficient meta-training while significantly improving generalization capabilities. Experimental results, obtained across various simulated tasks of the MuJoCo, Meta-World and ManiSkill benchmarks, indicate a significant improvement in learning efficiency and adaptability compared to the state-of-the-art. Our approach enhances the agent's ability to generalize from limited data and paves the way for more robust and versatile AI systems.

## 1 INTRODUCTION

Reinforcement learning (RL) (RL; Sutton & Barto, 2018) has made rapid progress in recent years and has shown success in complex real-world benchmarks (Bellemare et al., 2020; Degrave et al., 2022; Kaufmann et al., 2023) that go beyond typical (video) game playing benchmarks (Mnih et al., 2015a; Schrittwieser et al., 2020). Despite these high-profile success stories, RL has not yet found widespread adoption in many real-world applications. This is partly due to the high computational demands (Cobbe et al., 2020; Shala et al., 2022; 2023a) and the brittleness of RL algorithms (Henderson et al., 2018; Engstrom et al., 2020; Andrychowicz et al., 2021), which is further amplified by the difficulty of optimizing the hyperparameters of RL algorithms (Parker-Holder et al., 2022; Eimer et al., 2023; Mohan et al., 2023). More importantly, RL training pipelines are generally not built with generalization in mind (Kirk et al., 2023).

The lack of generalization capabilities is a crucial factor that limits the applicability of RL in domains where there are no perfect simulators. The classical RL training pipeline involves training, validating, and testing in the exact same task. However, in reality, an RL agent's sensors might slightly drift over time or tasks might exhibit drastically different features during different stages of an episode (such as, e.g., times of the day). If the training task was not set up with such changes in mind, an RL policy will not have the opportunity to learn generalizable behavior. Consequently, small variations in the task that might not have been observed during training can lead to severe failure cases. For example, a policy that was trained to steer a robot in a highly controlled lab environment might fail the instant it is supposed to leave the familiar lab environment.

Meta-learning (Vanschoren, 2019; Hospedales et al., 2022) is a subfield of machine learning that aims to take advantage of previous experience as efficiently as possible so that learned behavior can be transferred to similar problems. One promising approach within meta-learning is to include meta-features directly in the training procedure (Benjamins et al., 2023; Beukman et al., 2023; Shala et al., 2023b; Arango et al., 2024), enabling policies capable of zero-shot generalization based on observed meta-features. These meta-features are either hand-crafted (Benjamins et al., 2023; Beukman et al., 2023; Arango et al., 2024) or learned (Shala et al., 2023b). In the Meta-RL setting, it might not always be possible to have access to meaningful hand-crafted meta-features to learn generalizable policies. That is why, most work on meta-features for Meta-RL focuses on learned meta-features (Duan et al., 2016; Rakelly et al., 2019; Zintgraf et al., 2020; Melo, 2022; Grigsby et al., 2024a). These meta-features are generally learned through the use of recurrent neural networks (RNNs)

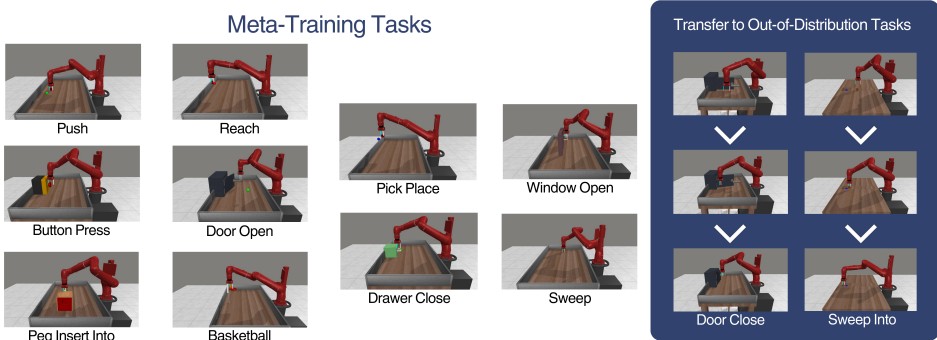

Figure 1: Illustration of the setting of the ML10 Benchmark in Meta-World. We evaluate ECET in each task and collect these frames. On the left are the tasks ECET is trained on, whereas on the right we show 3 frames (*beginning*, *middle* and *end* of the episode) from evaluating on 2 of the 5 test tasks. Though these tasks are not present in the training set, ECET manages to generalize and successfully close the door and sweep the block into the hole.

(Duan et al., 2016; Zintgraf et al., 2020), amortized inference (Rakelly et al., 2019) or transformers (Melo, 2022; Grigsby et al., 2024a). For instance, Rakelly et al. (2019) infer a posterior over meta-features by sampling transitions from the same task without explicitly considering the order of sequencing of these transitions. Thus, the meta-features generated have limited information on the task dynamics. On the other hand, RNN- and transformer-based methods process entire sequences of recent transitions spanning a few episodes to infer meta-features. This enables such methods to also incorporate dynamics information in the meta-features. Due to the vanishing gradient problem and the limited memory capacity of RNNs, the meta-features generated by RNN-based methods often emphasize characteristics of recent transitions, focusing primarily on intra-episode experiences. Similarly, Melo (2022) propose TrMRL, a transformer-based Meta-RL approach that generates meta-features by taking as input a sequence of only $T = 5$ most recent transitions. Grigsby et al. (2024a) with AMAGO further increase the amount of task information processed to generate meta-features by utilizing a transformer architecture capable of processing longer sequences of transitions spanning multiple episodes (i.e., cross-episode experiences). While exhibiting impressive performance in Meta-RL, both TrMRL and AMAGO present substantial computational overhead with a runtime complexity depending quadratically on the complete sequence length processed.

In this paper, we present a novel method for learning meta-features through efficient cross-episode[1] Meta-RL using a novel transformer architecture. Our approach enhances the adaptability of RL agents across diverse sets of tasks by leveraging both intra-episode and cross-episode experiences, providing a more comprehensive learning process. Our learned algorithm is capable of outperforming the previous state-of-the-art and providing more efficient meta-training in terms of timesteps, while significantly improving generalization capabilities. To foster reproducibility, we provide our code at `https://github.com/machinelearningnuremberg/ECET.git`.

## 2  RELATED WORK

Meta-Reinforcement Learning (Meta-RL)[2] emerged as a framework aimed at making RL algorithms more sample-efficient by leveraging knowledge learned from previous tasks to speed up learning for the current task. Meta-RL optimizes a bilevel optimization problem during the *meta-training* phase. The inner loop typically consists of traditional RL algorithms, while the outer loop represents meta-learning algorithms that exploit information from the inner loop to improve performance across multiple tasks. Once the meta-training phase is complete, only the inner loop algorithm is used in

---

[1]Throughout this paper, we use the term *cross-episode* to describe interactions across multiple episodes, which is synonymous with *inter-episode*. This choice aligns with the terminology used in the title for consistency

[2]For a recent survey, see Beck et al. (2023).

the *adaptation* phase. Due to the meta-training phase, the inner loop algorithm can achieve good performance using fewer interactions with the new task compared to starting from scratch.

Finn et al. (2017) proposed the Model-Agnostic Meta-Learning (MAML) algorithm, which can be used with any policy gradient algorithm as the inner loop to learn an initialization for the policy to speed up learning on a new test task. Duan et al. (2016) introduced $RL^2$, an approach that learns a reinforcement learning algorithm using RNNs, which share the hidden state across episodes from the same task. $RL^2$ uses a policy gradient algorithm as the meta-learning algorithm in the outer loop, with learning during the adaptation phase occurring only within the RNN's dynamics, without gradient adaptation. Another approach, PEARL, proposed by Rakelly et al. (2019), uses an off-policy RL algorithm, Soft Actor Critic (SAC) (Haarnoja et al., 2018), in the outer loop, and augments the state with stochastic meta-features of the task in the inner loop. An MLP generates the parameters of the posterior distribution of task meta-features.

Zintgraf et al. (2020) proposed VariBAD, a variant of $RL^2$ for Bayes-adaptive deep RL, which uses VAEs to learn an approximate posterior distribution over task features to augment the state input to the policy. VariBAD uses Proximal Policy Optimization (PPO) (PPO; Schulman et al., 2017) as the outer loop algorithm, but it can be replaced by any policy gradient algorithm. $RL^3$ (Bhatia et al., 2023) builds on $RL^2$ by incorporating action-value estimates as additional input to the RNN, enhancing its capacity to capture task features.

Under the umbrella of meta-RL, there has also been work on augmenting the meta-training task set to improve generalization (Lee & Chung, 2021; Rimon et al., 2022). These methods can be used with meta-RL algorithms based on task meta-features to improve generalization.

The transformer architecture (Vaswani et al., 2017) has revolutionized the field of Natural Language Processing (NLP) and beyond. Its reliance on self-attention mechanisms enables it to process entire sequences of data simultaneously, enhancing efficiency and handling global dependencies in data compared to RNNs. Parisotto et al. (2020) explored the application of transformer architecture in RL settings, addressing issues such as data efficiency and overfitting. Subsequent works, including the Decision Transformer (Chen et al., 2021) treated RL as a sequence modeling problem, leveraging the scalability and simplicity of transformers to efficiently handle RL tasks, a departure from conventional RL methods that typically rely on value function estimation or policy gradients. Janner et al. (2021) extended this concept, showcasing how transformers can unify various RL tasks into a single sequence modeling framework, promising improvements in offline RL algorithms. Extensions such as MGDT (Lee et al., 2022) and Gato (Reed et al., 2022) showcased the flexibility of transformers in multi-task and multi-modal RL, though they required expert demonstrations or fine-tuning for unseen tasks. Offline RL works use datasets of collected trajectories and a supervised prediction loss to train the transformer. This means that the learning process is based on a fixed dataset of experiences and that the model does not interact with the task during training. Although effective for certain scenarios, this approach might not adapt well to dynamic or unpredictable tasks where new experiences differ significantly from the training data.

Despite their promise, stabilizing transformer training in RL has proven challenging (Mishra et al., 2018; Parisotto et al., 2020; Melo, 2022; Grigsby et al., 2024a). Recent work has proposed stabilizing training through changes in the architecture (Parisotto et al., 2020) or the weight initialization schema (Huang et al., 2020; Melo, 2022). We follow Huang et al. (2020) and Melo (2022) and stabilize transformer optimization through weight initialization.

Numerous approaches exist in the literature for hierarchical transformers for offline RL (Shang & Ryoo, 2021; Correia & Alexandre, 2022; Mao et al., 2023; Zhang et al., 2023). These works mostly focus on leveraging datasets of demonstrations to perform well in the same task-distribution as the demonstrations. In contrast, our focus lies in online RL and generalization across tasks with varying dimensions (e.g., parametric and task variations). By leveraging the sequencing of transitions and cross-episode information, we aim to efficiently meta-learn task representations that adapt dynamically.

Meta-feature extraction lies at the heart of our method. Approaches like VariBAD (based on RNNs) and PEARL (based on amortized inference) extract task-relevant information from transitions within a task, augmenting the state input with meta-features. These meta-features capture the nuances of task dynamics and improve policy performance. Expanding on this, TrMRL (Melo, 2022) demonstrated that transformers can generalize better by using a sequence of the most recent transitions,

capturing dependencies through self-attention. Similarly, AMAGO (Grigsby et al., 2024a) showed that extracting information from longer sequences, spanning multiple episodes, significantly enhances generalization. In AMAGO-2 (Grigsby et al., 2024b), the authors evaluate their approach more extensively in multi-task RL problems and showcase the generalization abilities to unseen parametric variations of tasks in the meta-training set (e.g. harder levels).

Building on these foundations, our approach integrates intra-episode (within a single episode) and cross-episode (across multiple episodes) information to extract richer meta-features. Using transformers, we analyze sequences of transitions to capture complex, multi-level patterns that identify the task dynamics. This design enables in-context learning, allowing our model to dynamically adjust its behavior based on the context provided by past experiences.

Our method achieves efficiency in terms of runtime complexity while capturing intricate patterns in dynamic and diverse task environments. By leveraging information from both intra- and cross-episode experiences, we improve generalization across tasks with varying dimensions. Compared to prior state-of-the-art methods, our approach demonstrates improved meta-training efficiency and generalization.

## 3  PRELIMINARIES

The Reinforcement Learning (RL) problem is typically formalized within the framework of Markov Decision Processes (MDPs), defined by a tuple $(S, A, P, R, \gamma)$. $S$ represents a finite set of states in the task. Each state $s \in S$ encapsulates all the information that describes the situation at a particular point in time. $A$ denotes a set of actions that can be executed in all states. $P \colon S \times A \times S \to [0, 1]$ is the state transition probability function that describes the dynamics of the MDP. More specifically, $P(s'|s, a)$ gives the probability of transitioning to state $s'$ when action $a$ is taken in state $s$. $R \colon S \times A \to \mathbb{R}$ is the reward function, with $R(s, a)$ giving the immediate reward received after taking action $a$ from state $s$. $\gamma \in [0, 1]$ is the discount factor that determines the difference in importance between future rewards and current rewards. For an agent interacting with the MDP, lower values of $\gamma$ mean that the agent will prioritize immediate rewards more strongly, while a higher value indicates a preference for long-term gains.

The goal in RL is to find an optimal policy $\pi^* \in \Pi$, with $\pi \colon S \to A$, that maximizes the expected discounted cumulative reward. This can be formally defined as $\pi^* \in \mathrm{argmax}_{\pi \in \Pi} J(\pi)$ with

$$J(\pi) = \mathbb{E}\left[\sum_{t=0}^{\infty} \gamma^t R(s_t, a_t) \mid s_0 = s, \pi\right]$$

Here, $J(\pi)$ denotes the expected cumulative reward when starting in $s_0 = s$, playing action $a_t$ in state $s_t$ as dictated by policy $\pi$, i.e., $a_t \sim \pi(s_t)$. Most modern RL approaches parameterize the policy as $\pi_\theta$, where the parameters $\theta$ are typically the weights of a neural network, and aim to find the parameters that satisfy the previous expression.

**Meta-Reinforcement Learning (Meta-RL):** Given a distribution of tasks $p(\mathcal{T})$, where each task $\tau \sim p(\mathcal{T})$ can be considered a distinct MDP with its own state space $S^\tau$, action space $A^\tau$, transition dynamics $P^\tau$ and reward function $R^\tau$, the objective of a meta-RL technique is to learn a reinforcement learning algorithm $\mathcal{L}_\phi$ that directly outputs the parameters $\theta$ of a reinforcement learning policy. The learned algorithm $\mathcal{L}$ is parameterized by $\phi$ such that $\mathcal{L}_\phi \colon \mathcal{T} \to \theta$ and aims to learn the parameters $\theta$ so $\pi_\theta$ can generalize across a set of tasks sampled from the distribution $p(\mathcal{T})$. To this end, the policy interacts with a series of tasks $\tau$, drawn from $p(\mathcal{T})$, and aims to maximize its performance across this task distribution. This process involves two key phases:

*Meta-Training and Meta-Testing Phases:* During meta-training, the agent learns a policy $\pi_\theta$ and an update rule $\mathcal{L}_\phi$ (parameterized by $\theta$ and $\phi$, respectively) that can quickly adapt to new tasks, resulting in a bilevel optimization problem. This further involves learning a generalizable representation or prior knowledge that is transferable across tasks. The outer objective in this phase is to optimize the parameters $\phi$ such that $\phi^* \in \mathrm{argmax}_\phi G(\phi)$ with $G(\phi) = \mathbb{E}_{\tau \sim p(\mathcal{T})}[J(\pi_\theta)|\mathcal{L}_\phi, t]$ whereas the inner objective involves finding the weights of the policy that maximize $\theta^* \in \mathrm{argmax}_\theta J(\pi_\theta)$. In the meta-testing (adaptation) phase, the agent encounters new tasks $\tau_{\text{new}}$ sampled from $p(\mathcal{T})$. Using $\pi_{\theta^*}$

and $\mathcal{L}_{\phi^*}$, the agent quickly adapts to each new task. The adaptation effectiveness is measured by the agent's performance on these new tasks after a limited number of learning steps or experiences.

# 4 METHOD

We illustrate our proposed approach, which we dub ECET (**E**fficient **C**ross-**E**pisodic **T**ransformers), in Figure 2. Our architecture consists of two distinct groups of transformer encoder blocks. The first one is the *Intra-Episodic Transformer* (`IET`) that encapsulates the sequence of transitions inside an episode and allows us to understand the transition dynamics within each episode. The second group of encoder blocks is the *Cross-Episodic Transformer* (`CET`) that encodes the set of episodes and captures the transition dynamics across episodes.

Our experience on a task consists of $E$ episodes, including the most recent episode. For efficiency reasons, we sample a sequence of $T$ transitions from each episode. For the current episode, we ensure that the sequence ends with the current state observation of the agent, such that it can choose the most appropriate next action. The set of $E$ episodes, each having $T$ transitions, are stored as sequences of tuples $\chi_t^e = (s_t^e, a_t^e, r_t^e, d_t^e)_{e \in \{1,\dots,E\}, t \in \{1,\dots,T\}}$ where $d_t^e \in \{0,1\}$ is a termination flag. The $t$-th transition from the sequence of the $e$-th episode is denoted as $\chi_t^e \in S \times A \times R \times \{0,1\}$, while a sequence of $T$ transitions from the $e$-th episode as $(\chi_0^e, \chi_1^e, \dots, \chi_T^e) \in (S \times A \times R \times \{0,1\})^T$.

The *Intra-Episodic Transformer* (`IET`) computes a representation that transforms a sequence of $T$ transitions from the $e$-th episode into a vector representation $z^e \in \mathbb{R}^K$ as formalized below:

$$\mathbf{z}^e = \text{IET}\left(\chi_1^e, \dots, \chi_T^e\right), \qquad \text{IET} : (S \times A \times R \times \{0,1\})^T \to \mathbb{R}^K, \qquad \mathbf{z}^e \in \mathbb{R}^K$$

The `IET` employs an encoder-only transformer architecture with positional encoding, that projects the sequence of transitions to a latent representation using multiple layers of transformer blocks. The network's representation of the sequence is the $\mathbb{R}^K$ latent embedding of the $T$-th transition in the last transformer block.

We then input the intra-episodic representations $\mathbf{z}^e \in \mathbb{R}^K, \forall e \in \{1, \dots, E\}$ from all the episodes into the *Cross-Episodic Transformer* (`CET`), which captures the dynamics of sequences from different experience episodes. The `CET` network ultimately represents the experience on a task into a vector $\mathbf{z}_{\text{task}}$, formalized as:

$$\mathbf{z}_{\text{task}} = \text{CET}\left(\mathbf{z}^1, \dots, \mathbf{z}^E\right), \qquad \text{CET} : \mathbb{R}^{E \times K} \to \mathbb{R}^D, \qquad \mathbf{z}_{\text{task}} \in \mathbb{R}^D$$

The `CET` network is also a stack of transformer blocks. The representation of the episodic experience $\mathbf{z}_{\text{task}}$ is the $\mathbb{R}^D$ output for the $E$-th element of the last transformer block in `CET`. Since episodes $\mathbf{z}^e \in \mathbb{R}^K, \forall e \in \{1, \dots, E\}$ represent a set, we construct `CET` without positional encoding.

Finally, we condition the policy on the learned cross-episodic features and a linear transformation $\varphi$ of the current state $s$:

$$\lambda = \pi(\varphi(s), \mathbf{z}_{task})$$

where the output of the policy, $\lambda$, represents the parameters of the distribution $p_a$ from which we sample the next action $a \sim p_a(\lambda)$. We use a Gaussian distribution as the model for the probability distribution of the actions, therefore, $\lambda$ is the mean and variance of the distribution.

Following the RL$^2$ framework, we use PPO as the outer-loop algorithm to train our pipeline. We use the PPO objective for the actor and Negative Log-Likelihood loss for the critic. We detail our Meta-Training and Meta-Testing Algorithms in Algorithm 2 and Algorithm 3 respectively.

The hierarchical nature of our novel transformer architecture improves the runtime complexity of computing the task representation. Suppose we would have opted for a standard (non-hierarchical) transformer network (Melo, 2022) that is fed a single long sequence of all the transitions from all episodes (i.e. $E \cdot T$ many transitions). Such a naive approach requires an attention matrix of complexity $O(E^2 T^2)$. On the other hand, using the `IET` network to only capture intra-episodic features requires an attention matrix of complexity $O(T^2)$. Capturing cross-episodic features $\mathbf{z}_{\text{task}}$ using `CET` requires an attention matrix of complexity $O(E^2)$. Thus, our two-tiered architecture of the cross-episodic transformer has a runtime complexity of $O(E^2 + T^2)$, and scales to longer sequences and more episodes than a standard non-hierarchical transformer.

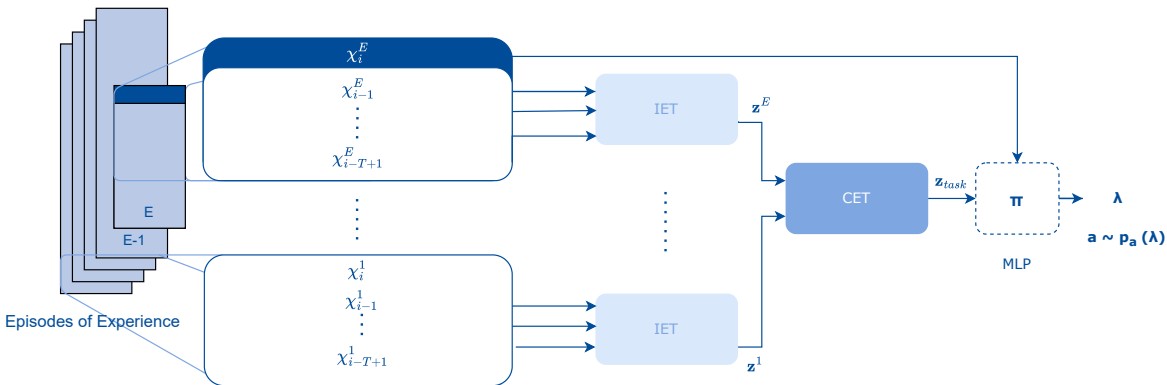

Figure 2: Illustration of our ECET architecture. Transitions $\chi_t^e = (s_t^e, a_t^e, r_t^e, d_t^e)$ represent which state $s_t^e$, action $a_t^e$, and reward $r_t^e$ were observed at the $t$-th transition of the $e$-th episode, but also a termination flag $d_t^e$ that indicates if the $e$-th episode already terminated at time $t$. By inputting the sequences independently through the Intra-Episodic Transformer (IET), we generate a feature vector $\mathbf{z}^e$ for each episode $e \in \{1, \ldots, E\}$. We then pass these sequence representations through the Cross-Episodic Transformer (CET) to generate a task representation $\mathbf{z}_{\text{task}}$ as input to the policy $\pi$.

# 5 EXPERIMENTS

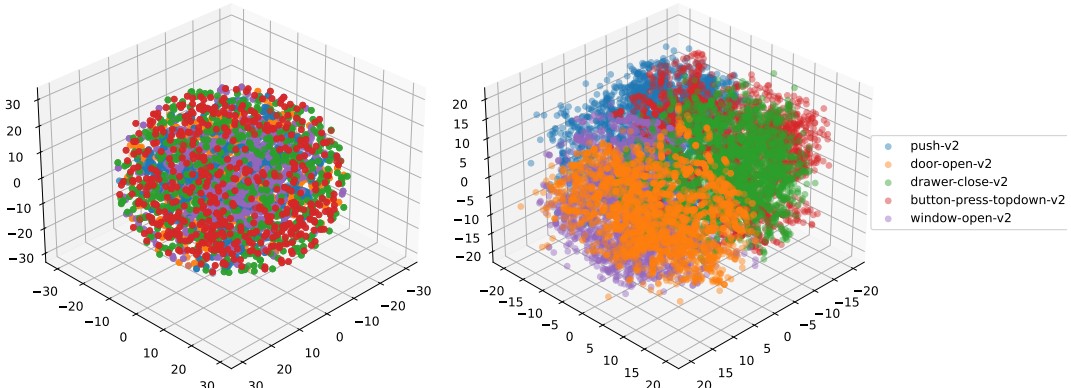

Figure 3: T-SNE plots of the output embeddings for TrMRL (left), and our ECET (right) for five of the tasks in the training set of the ML10 benchmark of Meta-World. For better visibility, we only plot 5 tasks here but provide a version with all 10 in Figure 14.

## 5.1 EXPERIMENTAL PROTOCOL

**MuJoCo**  We evaluate ECET and our baseline methods on MuJoCo (Todorov et al., 2012) locomotion tasks, which are widely used in the Meta-RL literature. Specifically, we consider the **AntDir** and **HalfCheetahDir** tasks, where the agent is required to move either forwards or backwards. Additionally, we include the **HalfCheetahVel** task, where the agent must adapt to running at different target velocities. The maximum episode length for these tasks is 200 timesteps.

**Meta-World**  To assess the performance of our cross-episode transformer approach, we use the Meta-World Benchmark (Yu et al., 2019). This benchmark is designed to help evaluate the performance of meta-RL baselines in 50 different simulated robotic manipulation tasks. These tasks all share the same state $S$ and action space $A$ as they are all located on a tabletop environment with a Sawyer arm (see Appendix A.4 for details and illustrations of all tasks). What differs in each task is the

behavior required to complete it successfully. For meta-learning, Meta-World includes three modes with varying degrees of difficulty (for details, see Appendix A.4):

- **ML1** is designed to evaluate few-shot adaptation to parametric variations within one task.
- **ML10** evaluates few-shot adaptation to 5 unseen test tasks, after training on 10 tasks.
- **ML45** is similar to, but more complex than ML10, with 45 training and 5 test tasks.

For ML1, the desired goal position is not provided as input, so the meta-RL algorithm needs to determine the location of the goal through trial and error. Similarly, for ML10 and ML45, task IDs are not provided as input, so the meta-RL algorithm needs to identify tasks from experience. The maximum episode length for all MetaWorld tasks is 500 timesteps.

**ManiSkill** We additionally use the ManiSkill Benchmark (Gu et al., 2023) to assess the performance of our proposed method on tasks where the state representation is an image. This benchmark consists of robotic manipulation tasks to help evaluate the performance of approaches for embodied AI in terms of their generalization to parametric variations, as well as variations in the object being manipulated. We evaluate on the *StackCube* and *PickSingleYCB* tasks, located on a tabletop environment with a Panda robot. What differs in each task is the behavior required to complete it successfully:

- **StackCube**: The goal is to pick up a red cube and stack it on top of a green cube and let go of the cube without it falling. The task is parameterized by randomly sampling the initial position of the cubes.
- **PickSingleYCB**: The goal is to pick up a random object sampled from the YCB dataset (Calli et al., 2015) and move it to a random goal position. This task is designed to evaluate few-shot adaptation to different objects.

We use the CNN architecture from Mnih et al. (2015b) to process RGB states for all methods. The maximum episode length for all ManiSkill tasks is 50 timesteps.

**Performance metrics** For the MuJoCo benchmark, we use the *average return* as a performance metric. The authors of Meta-World and ManiSkill propose the *success rate* as a performance evaluation metric. The distance of the task-relevant object from the goal position determines whether the agent is successful or not in an episode. Thus, the success rate is the fraction of episodes where the agent manages to place the task-relevant object inside a pre-set perimeter around the goal position.

Due to the fact that the tasks have different levels of difficulty, especially for the ML10 and ML45 modes in Meta-World, we additionally use the *average rank* as a performance metric for aggregating the performance across different tasks. We calculate the rank of methods over time for each task in the set based on their success rate, and then average over those ranks across tasks.

**Baselines:** We focus on investigating the performance of ECET compared to other online meta-reinforcement learning methods, namely MAML PPO (Finn et al., 2017), $RL^2$ (Duan et al., 2016), PEARL (Rakelly et al., 2019), VariBAD (Zintgraf et al., 2020), TrMRL (Melo, 2022) and AM-AGO (Grigsby et al., 2024a). Detailed descriptions of each are given in Appendix A.5. We conducted all experiments on a compute cluster of NVIDIA A100 GPUs. We trained all methods for $10^7$ steps in the MuJoCo environments, $5 \times 10^7$ steps in Meta-World, and $2.5 \times 10^7$ steps in ManiSkill. We repeat each run for 5 different random seeds. When comparing the methods in the figures, we plot the mean and standard error of the performance metrics across seeds.

### 5.2 HYPOTHESES AND RESULTS

*Hypothesis 1:* **ECET captures more general task features compared to state-of-the-art meta-RL methods. This stems from ECET's ability to capture intra- as well as cross-episodic experiences.** We compare the embeddings that ECET learns to those of TrMRL to investigate the extent to which they differentiate between tasks. We sample episodes from the training tasks in ML10 and plot the output of the transformers for both ECET and TrMRL through dimensionality reduction. For the sake of visual clarity, in Figure 3 we show the results for 5 of these tasks. For the plot with all the 10 tasks, see Figure 14. Additionally, in Figure 15 we compare the intra- and inter-task embedding distances. From these plots it is visible that the learned embeddings

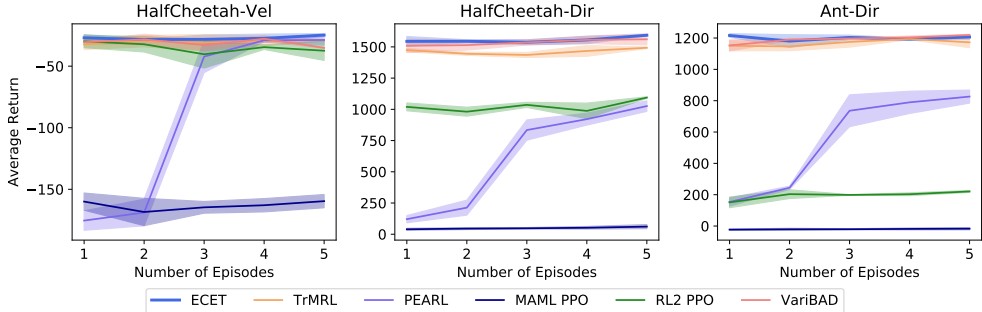

Figure 4: Test performance in terms of average return across rollouts for ECET, TrMRL, PEARL, MAML PPO, RL2 PPO and VariBAD on the *HalfCheetah-Vel*, *HalfCheetah-Dir*, and *Ant-Dir* tasks of the MuJoCo benchmark, testing adaptation to parametric variations of these tasks. We show the corresponding meta-training performance plot in Figure 12.

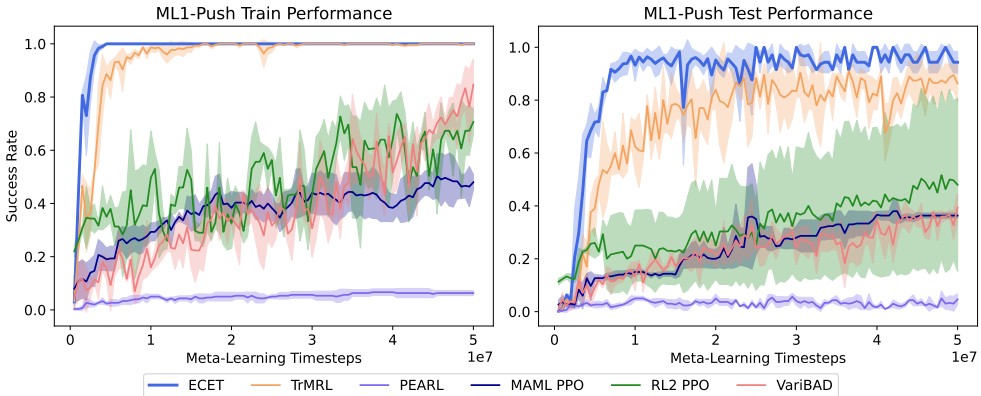

Figure 5: Meta-Train and Test performance in terms of average success rate for ECET, TrMRL, PEARL, MAML PPO, RL2 PPO and VariBAD on the ML1 benchmark for training(left) and testing(right) on parametric variations of the *Push* task. We show the corresponding plot with respect to time in Figure 19, whereas the average rank plot can be found in Figure 16.

of ECET produce a clearer grouping for the same task, compared to the embeddings of TrMRL. Furthermore, we can see a grouping of tasks that are similar to each other, e.g., *Window Open* and *Door Open*. This shows the usefulness of cross-episodic experiences in task differentiation. For easier analysis, we provide scripts for interactive versions of the T-SNE plots in our code repository `https://github.com/machinelearningnuremberg/ECET/blob/main/plotly_interactive_plot.py`.

*Hypothesis 2:* **ECET outperforms the state-of-the-art in online meta-RL algorithms in few-shot adaptation to parametric variations of tasks.** To evaluate adaptibility to parametric variations of the task, we compare ECET and the baselines in the MuJoCo set of tasks, and in the ML1 mode of Meta-World (*Push* and *Reach* tasks). We evaluate the generalization on the parametrized MuJoCo tasks through doing 5 rollouts at test time, and calculating the return for each. We show the results in Figure 4. ECET, TrMRL, and VariBAD show robust test performance across the three tasks. For the Meta-World ML1 benchmark, we show the performance in terms of Success Rate in Figure 5 for *Push*. The figure contains results for meta-training (left) and test (right) performance in the case where we train on variations of the *Push* task, and test the adaptability of the methods on a set of unseen variations of the same task. In terms of meta-training performance (left), we can see that transformer-based methods are faster in differentiating between the variations of the task and thus achieve a maximum success rate of 1.0. In terms of test performance (right), ECET and TrMRL generalize successfully, reaching a test success rate greater than 0.8. In addition, ECET shows a

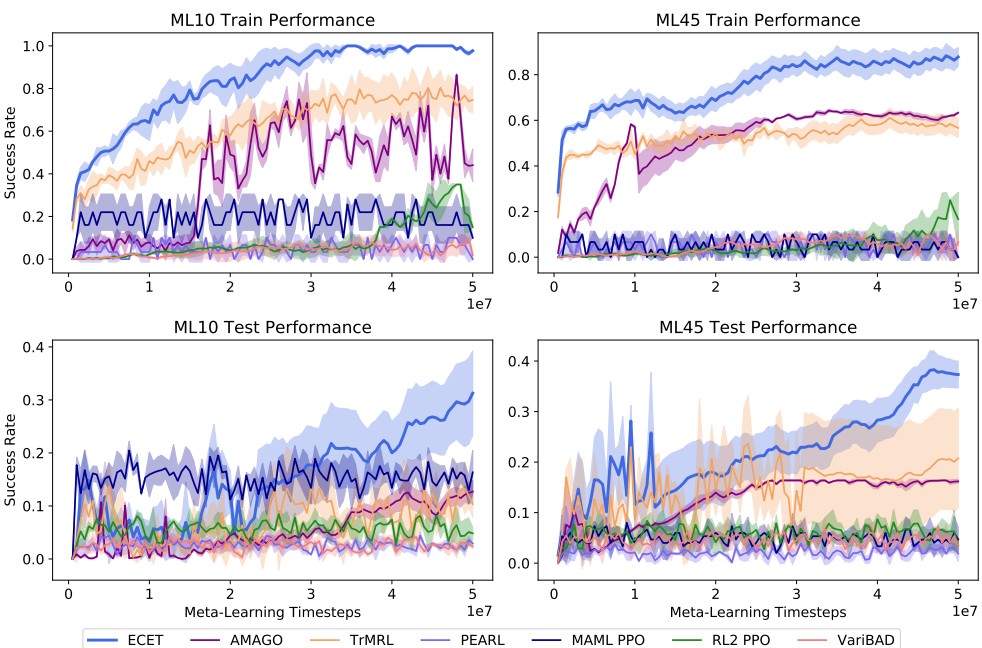

Figure 6: Meta-Train and Test performance in terms of average success rate for ECET, AMAGO, TrMRL, PEARL, MAML PPO, RL2 PPO and VariBAD on the ML10(left) and ML45(right) benchmarks. Here, we are training and testing on disjoint sets of tasks. We show the corresponding plot with respect to time in Figure 21, whereas the average rank plot can be found in Figure 17.

higher generalizability compared to TrMRL. RL2 PPO struggles to robustly generalize, exhibiting a higher variation in performance compared to the other methods. In Figure 13 in Appendix A.6 we show the Meta-Train and Test performance on variations of the *Reach* task. The task variability for *Reach* is not as difficult to learn, thus all the methods perform better both when meta-training and testing. To further assess generalization to parametric variation, we evaluate our method and the baselines in the *StackCube* task of ManiSkill. We show the meta-train and test performance in Figure 7 (left column), where we notice a similar pattern on performances as with ML1, with ECET outperforming the baselines after $2.5 \cdot 10^7$ meta-training timesteps.

*Hypothesis 3:* **ECET outperforms the state-of-the-art online meta-RL in out-of-distribution (OOD) tasks.** We define out-of-distribution (OOD) performance to be the performance of a meta-learned agent on a set of tasks that is disjoint from the tasks it is meta-learned on. We show the performance of the methods in terms of the average success rate for the meta-training (top) and test (bottom) performance for ML10 (left) and ML45 (right) in Figure 6. In this setting we also compare to AMAGO, as a recently proposed powerful transformer-based method processing long sequences. ECET is the only method that achieves an average success rate greater than $0.8$ in ML10 and greater than $0.6$ for ML45 in terms of meta-training performance with a final success rate roughly $20\%$ higher than the baselines. The other methods fail to learn policies that successfully identify the different tasks in the training set for the given budget of $5 \times 10^7$ meta-learning timesteps, and thus struggle to solve them. In terms of OOD performance, the bottom row shows that generalization to distinct tasks is still a challenge for meta-RL. ECET achieves an average test success rate of approximately $0.35$ for ML10 and $0.4$ for ML45. Similarly, in Figure 7 (right column) where we show the performance on *PickSingleYCB*, we notice that all baselines except for TrMRL struggle to effectively generalize on the task of picking different objects. ECET is the best performing method, achieving an average test success rate of $0.39$. This further demonstrates that cross-episodic features can improve meta-RL performance without incurring the cost of gradient adaptation at test time.

To focus on comparing the methods based on their robustness in relative performance to each other, in Appendix A.6.2 we provide plots showing the average ranks of the methods as a performance metric. These plots show that ECET robustly achieves the highest relative performance (and thus the

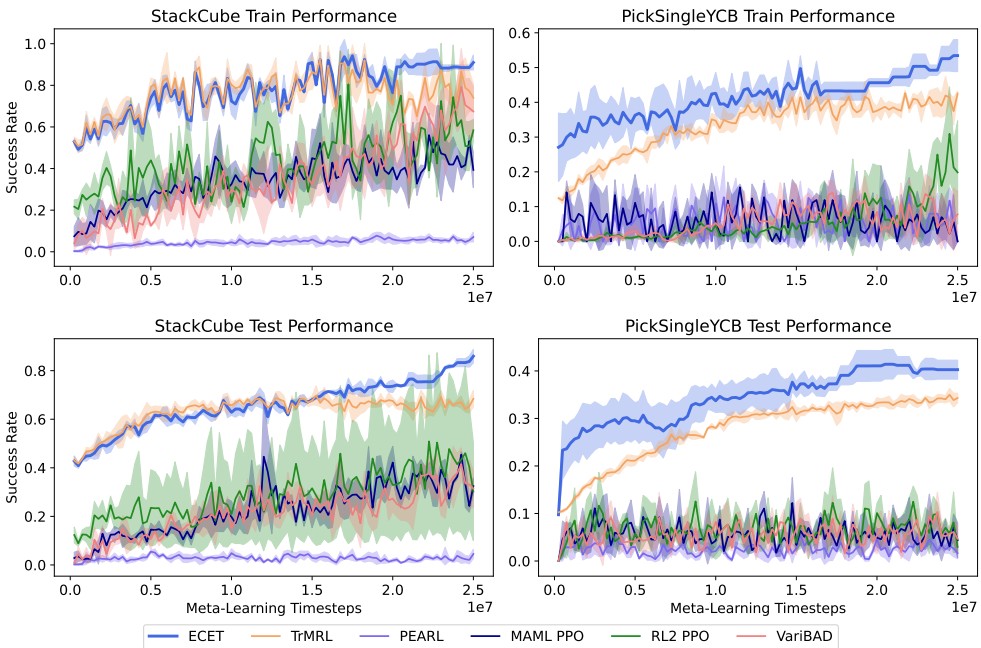

Figure 7: Average success rate plots of Meta-Train and Test performance for all methods on the *StackCube*(left) and *PickSingleYCB*(right) tasks. We show the corresponding plot with respect to time in Figure 22, whereas the average rank plot can be found in Figure 18.

lowest rank) across the meta-learning benchmarks we consider. For a detailed illustration of per-task performance, we refer to Appendix A.6.6. In Appendix A.6.4 through plots of episodic return across 50 episodes at test time, we show that ECET performs well even in scenarios when the agent has not yet collected the $E$ episodes of experience.

## 5.3 ABLATIONS

We investigate the behavior and the impact of ECET's components on performance in Appendix A.6.5. We show the effect of varying the number of episodes ($E$) in memory and the sequence length ($T$) of sampled transitions in Figure 28. Figure 29 (top plot) shows that sampling without replacement results in more reliable and faster performance improvements, especially for smaller ($E$) and ($T$). In terms of architectural variations, we investigated the effects of adding positional encoding in the Cross-Episode Transformer (CET) and simplifying ECET to a flat architecture akin to TrMRL and AMAGO. The results in Figure 26 indicate that positional encoding in CET introduces misleading ordering, leading to suboptimal convergence, while the flat architecture struggles to capture comprehensive task representations.

## 6 CONCLUSION

We introduced Efficient Cross-Episode Transformers (ECET) for Meta-Reinforcement Learning, a powerful approach to online meta-RL. Our experiments within the MuJoCo, Meta-World and ManiSkill benchmarks highlight our method's capability for few-shot adaptation to tasks with parametric variations, as well as its capacity to adjust to completely unseen tasks that share the same state and action spaces as the training tasks. Despite outperforming state-of-the-art methods in online meta-RL by efficiently compressing cross-episodic knowledge, our method's generalization to tasks that are completely different to the meta-training set remains limited as our approach is tailored to exploiting similarities. Future work should consider using task augmentation to improve generalization, modifying the adaptation phase, or increasing model capacity to handle a wider context in order to improve performance in this phase.

## ACKNOWLEDGEMENTS

Gresa Shala, André Biedenkapp and Josif Grabocka acknowledge funding by The Carl Zeiss Foundation through the research network "Responsive and Scalable Learning for Robots Assisting Humans" (ReScaLe) of the University of Freiburg. The authors gratefully acknowledge the scientific support and HPC resources provided by the Erlangen National High Performance Computing Center (NHR@FAU) of the Friedrich-Alexander-Universität Erlangen-Nürnberg (FAU). The hardware is funded by the German Research Foundation (DFG).

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

# A  APPENDIX

## A.1  RL$^2$ FRAMEWORK

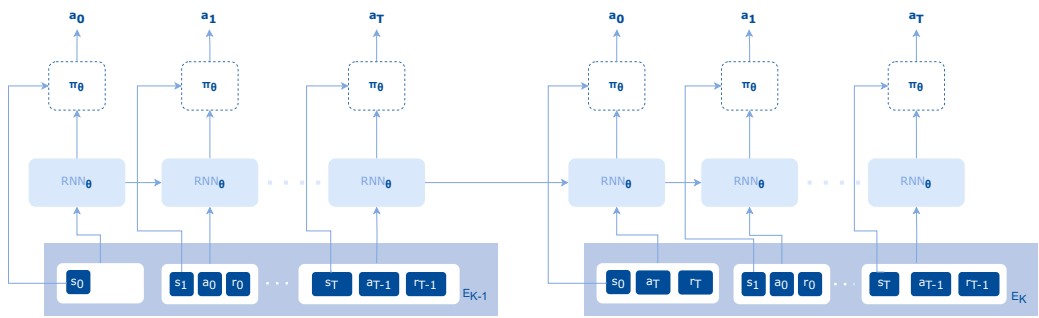

Figure 8: Illustration of the RL$^2$ framework, based on Figure 4 from Beck et al. (2023). As illustrated, the RNN takes as input the current state $s_t$, the action(if there is any) that lead to the state and the reward incurred from reaching that state. The output of the RNN is then used as input to the policy $\pi$ along with the current state.

## A.2  META-TRAINING AND META-TESTING ALGORITHMS

We include the detailed algorithm for Meta-Training in Algorithm 2 and the algorithm for Meta-Testing in Algorithm 3. In order to be consistent in comparison to the baselines, we use the garage contributors (2019) library implementations for the PPO algorithm, Gaussian Actor and Gaussian Critic.

### A.2.1  INITIAL SETUP AND HYPERPARAMETERS OF META-TRAINING

We use as input a distribution of meta-training tasks $\mathcal{T}$ and a history $\mathcal{H}$, a FIFO queue that stores past transitions to capture the temporal dynamics. The hyperparameters that guide the meta-training process include: the number of episodes $E$ to maintain in the history for context, the sequence length $S$ of transitions to be processed by the Intra-Episode Feature Extractor **IET**, the maximum number of training epochs $max\_epochs$, the sizes for both meta and mini-batches ($meta\_batch\_size$, $mini\_batch\_size$), and the number of episodes $k$ collected per task in each training round.

### A.2.2  PARAMETER DEFINITIONS

**Actor Parameters** include the parameters for embedding transitions ($\varphi_1$) and states ($\varphi_2$), the parameters for our Intra- and Inter-Episode Feature Extractors ($\Theta_1, \Theta_2$) within the hierarchical transformer framework, and the policy network ($\pi$) parameters ($\Theta_3$). **Critic Parameters** include the parameters ($\Psi$) for our value function ($V$), which assists in estimating the expected returns and guiding the actor's policy adjustments.

### A.2.3  PROCEDURAL COMPONENTS

For clarity and conciseness of writing, we include some procedural components of our meta-training and meta-testing pipeline as procedures in Algorithm 1. These include:

- *CollectEpisode*: We gather one episode of interactions for a given task, utilizing the Actor ($\mathbf{\Phi}, \mathbf{\Theta}$) for action decisions, and record these transitions within our history $\mathcal{H}$ and an episode_buffer which contains for each transition in the episode: the set of sampled transition sequences $\mathcal{C}$ used to generate the task representation ,as well as the ($state, action, reward, terminated$) for processing.

- *CollectAndProcessEpisode*:  Using  the  episode_buffer  data  returned,  we  process  each  transition  to  compute  advantages  and  returns,  and  we  add

---

**Algorithm 1** Subroutines for ECET

---

**Input**: Distribution of tasks $\mathcal{T}$, history $\mathcal{H}$ (FIFO queue) to store past transitions.

**Hyperparameters**: number of episodes $E$ to keep in history, sequence length $T$ of transitions to input in **IET**, $max\_epochs$, $meta\_batch\_size$, $mini\_batch\_size$, number of episodes $k$ to collect for each task during training.

**Actor Parameters**

Transition($\varphi_1$) and state($\varphi_1$) embedding parameters: $\boldsymbol{\Phi} = \varphi_1, \varphi_2$

Intra-Episodic Transformer($\textbf{IET}(\cdot; \Theta_1)$), Cross-Episodic Transformer($\textbf{CET}(\cdot; \Theta_2)$), and policy($\pi(\cdot; \Theta_3)$) parameters: $\boldsymbol{\Theta} = \{\Theta_1, \Theta_2, \Theta_3\}$

**Critic Parameters**

Value function ($(V(\cdot; \Psi))$) parameters: $\Psi$

**procedure** COLLECTEPISODE($task, \boldsymbol{\Phi}, \boldsymbol{\Theta}, \mathcal{H}$)

    episode_buffer.clear()

    state ← task.reset()

    terminated ← False

    **while** not terminated **do**

        $\mathcal{C}$ ← SAMPLETRANSITIONS($\mathcal{H}, state$)

        $\mathbf{z}_{task}$ ← GENERATETASKREPRESENTATION($\mathcal{C}, \varphi_1, \Theta_1, \Theta_2$)

        $\lambda$ ← $\pi(\mathbf{z}_{task}, \varphi_2(state); \Theta_3)$

        action ∼ $\lambda$

        next_state, reward, terminated ← task.act(action)

        Add $(state, action, reward, terminated)$ to $\mathcal{H}$

        Add $(\mathcal{C}, state, action, reward, terminated)$ to episode_buffer

        state ← next_state

    **return** episode_buffer, $\mathcal{H}$

**procedure** COLLECTANDPROCESSEPISODE($task, \boldsymbol{\Phi}, \boldsymbol{\Theta}, \Psi, \mathcal{H}, \mathcal{D}$)

    episode_buffer, $\mathcal{H}$ ← COLLECTEPISODE($task, \boldsymbol{\Phi}, \boldsymbol{\Theta}, \mathcal{H}$)

    **for** $(\mathcal{C}, state, action, reward, terminated)$ in episode_buffer **do**

        Calculate $advantage$ using the value function $V(\cdot; \Psi)$, and received reward.

        Calculate the $return$.

        Add $(\mathcal{C}, state, action, reward, terminated, advantage, return)$ to $\mathcal{D}$

    **return** $\mathcal{D}, \mathcal{H}$

**procedure** SAMPLETRANSITIONS($\mathcal{H}, current\_state$)

    Initialize episode transitions set $\mathcal{C}$ to empty.

    **for** $episode \leftarrow 1$ to $E$ **do**

        **if** episode not current **then**

            Sample sequence of $T$ transitions and add to $\mathcal{C}$.

        **else**

            Sample sequence of latest $T$ transitions including $current\_state$ and add to $\mathcal{C}$.

    **return** $\mathcal{C}$

**procedure** GENERATETASKREPRESENTATION($\mathcal{C}, \varphi_1, \Theta_1, \Theta_2$)

    Initialize set $\mathcal{C}_{seq}$ to empty.

    **for** $i \leftarrow 1$ to $E$ **do**

        $T_i \leftarrow \mathcal{C}[i]$

        $\mathbf{z}_{seq}^i \leftarrow \textbf{IET}(\varphi_1(T_i); \Theta_1)$

        Add $\mathbf{z}_{seq}^i$ to $\mathcal{C}_{seq}$.

    $\mathbf{z}_{task} \leftarrow \textbf{CET}(\mathcal{C}_{seq}; \Theta_2)$.

    **return** $\mathbf{z}_{task}$

---

---

**Algorithm 2** Meta-Training with ECET using Subroutines from Algorithm 1

---

**Input**: Distribution of meta-training tasks $\mathcal{T}$, history $\mathcal{H}$ (FIFO queue) to store past transitions.

**Hyperparameters**: number of episodes $E$ to keep in history, sequence length $T$ of transitions to input in **IET**, $max\_epochs$, $meta\_batch\_size$, $mini\_batch\_size$, number of episodes $k$ to collect for each task during training.

**Actor Parameters**

Transition($\varphi_1$) and state($\varphi_1$) embedding parameters: $\mathbf{\Phi} = \{\varphi_1, \varphi_2\}$

Intra-Episodic Transformer(**IET**$(\cdot; \Theta_1)$), Cross-Episodic Transformer(**CET**$(\cdot; \Theta_2)$), and policy($\pi(\cdot; \Theta_3)$) parameters: $\mathbf{\Theta} = \{\Theta_1, \Theta_2, \Theta_3\}$

**Critic Parameters**

Value function $((V(\cdot; \Psi)))$ parameters: $\Psi$

**Procedures**: COLLECTANDPROCESSEPISODE, SAMPLETRANSITIONS

**procedure** ACTOR($\{(\mathcal{C}, state)\}_{i=1}^{mini\_batch\_size}, \mathbf{\Phi}, \mathbf{\Theta}$)

    Generate $\{z_{task}\}_{i=1}^{mini\_batch\_size}$ using GENERATETASKREPRESENTA-TION($\{\mathcal{C}\}_{i=1}^{mini\_batch\_size}, \varphi_1, \Theta_1, \Theta_2$).

    $\{\mu_{actor}, \sigma_{actor}\}_{i=1}^{mini\_batch\_size} \leftarrow \pi(\{z_{task}, \varphi_2(state)\}_{i=1}^{mini\_batch\_size}, ; \Theta_3)$.

    **return** $\{\mu_{actor}, \sigma_{actor}\}_{i=1}^{mini\_batch\_size}$

**procedure** UPDATEMODELS($\mathcal{D}, \mathbf{\Phi}, \mathbf{\Theta}, \Psi$)

    $\mathbf{\Phi}^{old}, \mathbf{\Theta}^{old} \leftarrow \mathbf{\Phi}, \mathbf{\Theta}$

    **for** $\{(\mathcal{C}, state, action, reward, terminated, advantage, return)\}_{i=1}^{mini\_batch\_size}$ in $\mathcal{D}$ **do**

        $\lambda^{old} \leftarrow$ ACTOR($\{(\mathcal{C}, state)_i\}_{i=1}^{mini\_batch\_size}, \mathbf{\Phi}^{old}, \mathbf{\Theta}^{old}$).

        $prob^{old} \leftarrow \lambda^{old}(\{action_i\}_{i=1}^{mini\_batch\_size})$.

        $\lambda \leftarrow$ ACTOR($\{(\mathcal{C}, state)_i\}_{i=1}^{mini\_batch\_size}, \mathbf{\Phi}, \mathbf{\Theta}$).

        $prob \leftarrow \lambda(\{action_i\}_{i=1}^{mini\_batch\_size})$.

        $advantages \leftarrow \{advantage_i\}_{i=1}^{mini\_batch\_size}$

        Calculate the PPO actor loss with respect to $\mathbf{\Phi}, \mathbf{\Theta}$:

$$L^{actor}(\mathbf{\Phi}, \mathbf{\Theta}) = \hat{\mathbb{E}}_t\left[\min(r_t(\mathbf{\Phi}, \mathbf{\Theta})advantages, \text{clip}(r_t(\mathbf{\Phi}, \mathbf{\Theta}), 1 - \epsilon, 1 + \epsilon)advantages)\right]$$

        where $r_t(\mathbf{\Phi}, \mathbf{\Theta})$ is the probability ratio $r_t(\mathbf{\Phi}, \mathbf{\Theta}) = \frac{prob}{prob^{old}}$.

        $\mu_{critic}, \sigma_{critic} \leftarrow V(\{state_i\}_{i=1}^{mini\_batch\_size})$

        $returns \leftarrow \{return_i\}_{i=1}^{mini\_batch\_size}$

        Calculate the NLL for the critic with respect to $\Psi$:

$$L^{critic}(\Psi) = -\frac{1}{N}\sum_{i=1}^{N}\log\left(\frac{1}{\sigma_{V_i}\sqrt{2\pi}}\exp\left(-\frac{(returns - \mu_{critic})^2}{2\sigma_{critic}^2}\right)\right)$$

        Update $\mathbf{\Phi}, \mathbf{\Theta}$ and $\Psi$ by minimizing $L^{actor}$ and $L^{critic}$.

Initialize $\mathbf{\Phi}, \mathbf{\Theta}, \Psi$.

**for** epoch $= 1$ to $max\_epochs$ **do**

    Empty meta-training dataset $\mathcal{D}$.

    Sample $meta\_batch\_size$ tasks $\tau$ from $\mathcal{T}$.

    **for** each sampled $task$ **do**

        Initialize $\mathcal{H}$ to zeros.                     ▷ FIFO queue of $E$ episodes

        **for** $episode \leftarrow 1$ to $k$ **do**

            $\mathcal{D}, \mathcal{H} \leftarrow$ COLLECTANDPROCESSEPISODE($task, \mathbf{\Phi}, \mathbf{\Theta}, \Psi, \mathcal{H}, \mathcal{D}$)

    UPDATEMODELS($\mathcal{D}, \mathbf{\Phi}, \mathbf{\Theta}, \Psi$)

---

---

**Algorithm 3** Meta-Testing with ECET using Subroutines from Algorithm 1

---

**Input**: Distribution of meta-testing tasks $\mathcal{T}_{test}$, history $\mathcal{H}$ (FIFO queue) to store past transitions, $num\_eval\_tasks$, $num\_eval\_episodes$.

**Parameters** from meta-training

Transition($\varphi_1$) and state($\varphi_1$) embedding parameters: $\mathbf{\Phi} = \{\varphi_1, \varphi_2\}$

Intra-Episodic Transformer($\mathbf{IET}(\cdot; \Theta_1)$), Cross-Episodic Transformer($\mathbf{CET}(\cdot; \Theta_2)$), and policy($\pi(\cdot; \Theta_3)$) parameters: $\mathbf{\Theta} = \{\Theta_1, \Theta_2, \Theta_3\}$.

**Procedures**: COLLECTEPISODE

Sample $num\_eval\_tasks$ tasks from $\mathcal{T}_{test}$.

**for** each sampled $task$ **do**

    Initialize $\mathcal{H}$ to zeros.                        ▷ FIFO queue of $E$ episodes

    **for** $episode \leftarrow 1$ to $num\_eval\_episodes$ **do**

        $\_, \mathcal{H} \leftarrow$ COLLECTEPISODE($task, \mathbf{\Phi}, \mathbf{\Theta}, \mathcal{H}$)

---

$(\mathcal{C}, state, action, reward, terminated, advantages, returns)$ to the meta-training dataset $\mathcal{D}$. We return the updated history $\mathcal{H}$ and meta-training dataset $\mathcal{D}$.

- *SampleTransitions*: Using the history $\mathcal{H}$, we sample a sequence of $T$ transitions from each episode in $\mathcal{H}$, except for the current episode, where we sample a sequence of $T$ transitions that ends with the $current\_state$. We add these sequences in a set $\mathcal{C}$ and return it.

- *GenerateTaskRepresentation*: We first transform each transition in the sequences $T_i$ using the transition embedding $\varphi_1$, and then input the embedding into the Intra-Episode Feature Extractor to get $\mathbf{z}^i_{seq} \leftarrow \mathbf{IET}(\varphi_1(T_i); \Theta_1)$ for each episode $i$. We store the $\mathbf{z}^i_{seq}$ output embeddings from $\mathbf{IET}$ in the set $\mathcal{C}_{seq}$. We then input them in the Inter-Episode Feature Extractor to get $\mathbf{z}_{task} \leftarrow \mathbf{CET}(\mathcal{C}_{seq}; \Theta_2)$. $\mathbf{z}_{task}$ is the task representation we return.

### A.2.4 META-TRAINING

We describe the meta-training process in Algorithm 2, using the subroutines from Algorithm 1. We first initialize the Actor parameters ($\mathbf{\Phi}, \mathbf{\Theta}$) and Critic parameters ($\Psi$).

**Data Collection**. For each epoch until reaching $max\_epochs$, we start by initializing an empty meta-training dataset $\mathcal{D}$, and sampling $meta\_batch\_size$ number of tasks $\tau$ from $\mathcal{T}$. For each sampled task, we first initialize the history queue $\mathcal{H}$ to zeros. This queue has a length of $E$ episodes, to fit the hierarchical transformer's memory. For $k$ episodes we then update the meta-training dataset $\mathcal{D}$ and history $\mathcal{H}$ by calling the subroutine *CollectAndProcessEpisode*.

**Model Update**. Once we have collected data in $\mathcal{D}$ for $k$ episodes from each sampled task, we use it to update the parameters. As we also show in the subroutine *UpdateModels*, we copy the current $\mathbf{\Phi}$ and $\mathbf{\Theta}$ to $\mathbf{\Phi}^{old}$ and $\mathbf{\Theta}^{old}$, in order to be able to calculate the probability ratios for the PPO objective for the Actor. We sample a $mini\_batch\_size$ of tuples $(\mathcal{C}, state, action, reward, terminated, advantage, return)$ from $\mathcal{D}$. Using the $state$ values from the mini-batch, as well as the sampled transition sequences $\mathcal{C}$ for each of these states, we get outputs from the actor using $\mathbf{\Phi}^{old}$ and $\mathbf{\Theta}^{old}$. As we show in the *Actor* procedure, we use the mini-batch of sampled transition sequences $\mathcal{C}$ to generate a mini-batch of task representations $\mathbf{z}_{task}$ as described in the *GenerateTaskRepresentation* subroutine. We embed the mini-batch of states using the state embedding $\varphi_2$, and concatenate them with the task representation $\mathbf{z}_{task}$ for each state in the mini-batch. We then input the resulting mini-batch in the policy $\pi$ to get the mini-batch of parameters of the action distribution. Since the action distribution is a Gaussian, the policy $\pi$ outputs the mean $\mu_{actor}$ and standard deviation $\sigma_{actor}$. Since we use $\mathbf{\Phi}^{old}$ and $\mathbf{\Theta}^{old}$, we name this mini-batch of distribution parameters $\lambda^{old}$. We then calculate the probability of the actions in the mini-batch given the distribution parameters $\lambda_{old}$, and save the action probabilities in $prob^{old}$. We do the same procedure for $\mathbf{\Phi}$ and $\mathbf{\Theta}$, resulting with the mini-batch of action probabilities $prob$. Using these action probabilities, as well as the advantages from the mini-batch, we calculate the PPO Actor loss with respect to $\mathbf{\Phi}$ and $\mathbf{\Theta}$. As our Value function, similarly to the policy $\pi$ is also a Gaussian MLP, predicting the mean and standard deviation of a Gaussian distribution for the value of each state input, to calculate the Critic loss, we get the mini-batch of distribution parameters $\mu_{critic}$ and $\sigma_{critic}$ as output from the Value function V using the states in the minibatch as input. As the Critic loss, we calculate the Negative Log Likelihood loss of the returns in the mini-batch using $\mu_{critic}$ and $\sigma_{critic}$.

We update $\boldsymbol{\Phi}$ and $\boldsymbol{\Theta}$ by minimizing the PPO Actor Loss $\mathbf{L}^{actor}$. We update $\Psi$ by minimizing the NLL Critic Loss $\mathbf{L}^{critic}$.

We repeat this process until we go over all tuples in the meta-training dataset $\mathcal{D}$. Once $\mathcal{D}$ has been used for updates, we empty it, and return to sampling tasks and collecting episodes from them again, and updating the parameters based on the new dataset, and so on, until $max_epochs$ is reached.

### A.2.5 META-TESTING

We show the meta-test algorithm in Algorithm 3, using the *CollectEpisode* subroutine from Algorithm 1. We use a distribution of meta-testing tasks $\mathcal{T}_{test}$ and a FIFO queue as history $\mathcal{H}$ for the past episodes as input. We also set the $num\_eval\_tasks$ and $num\_eval\_episodes$ parameters which determine the number of tasks sampled from $\mathcal{T}_{test}$ and the number of episodes performed in each task, respectively. We use the Actor parameters $(\boldsymbol{\Phi}, \boldsymbol{\Theta})$ which we got from meta-training. We first sampled $num\_eval\_tasks$ from $\mathcal{T}_{test}$. For each sampled task, we initialize the history $\mathcal{H}$ to zeros, and then collect $num\_eval\_episodes$ and update $\mathcal{H}$ from each collected episode.

### A.3 ECET HYPERPARAMETERS

Table 1: Hyperparameters Across Benchmarks

| Hyperparameter | Benchmark | | |
| --- | --- | --- | --- |
| | MuJoCo | MetaWorld | Maniskill |
| Number of Transitions (T) | | 5 | |
| Number of Episodes (E) | 2 | 25 | |
| Number of Layers for IET | | 2 | |
| Number of Heads for IET | 16 | 4 | |
| Number of Layers for CET | | 2 | |
| Number of Heads for CET | 16 | 4 | |
| Minibatch Size | 256 | 32 | |
| Policy Learning Rate | 3e-5 | 5e-5 | |
| Critic Learning Rate | 3e-5 | 5e-5 | |

### A.4 BENCHMARK DETAILS

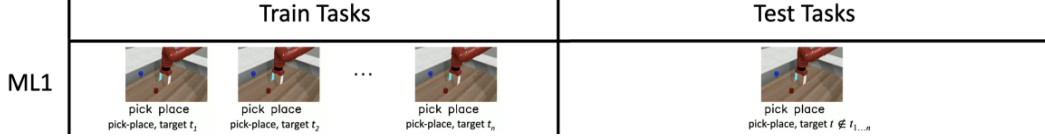

Figure 9: Meta-world Tasks in the M1 with training/test Split. Figure taken from (Yu et al., 2019)

**ML1**  provides one main task, with parametric variation in object and goal positions. ML1 enables evaluation in the few-shot adaptation setting. In particular, adaptations to parametric variations within one task. Figure 9 gives an example of meta-training and testing for the pick in place task, where only the test-task target position is different from those observed during training.

**ML10**  provides 10 different training tasks and 5 others to test the adaptation performance of meta-learning methods. This mode evaluates few-shot adaptation to 5 new test tasks, after training on 10 different tasks, see Figure 10. The test tasks are all related to the training tasks, but unseen operations have to be performed (such as opening a drawer at test time instead of closing it as learned in meta-training).

**ML45**  provides 45 highly different training tasks and 5 others to test the adaptation performance of meta-learning methods. This mode evaluates few-shot adaptation to 5 new test tasks, after training on 45 different tasks. Figure 11 shows all 50 tasks.

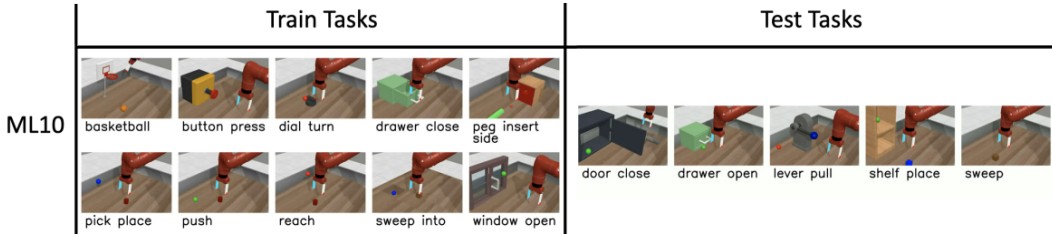

Figure 10: Meta-world Tasks in the M10 protocol with training/test Split. Figure taken from (Yu et al., 2019)

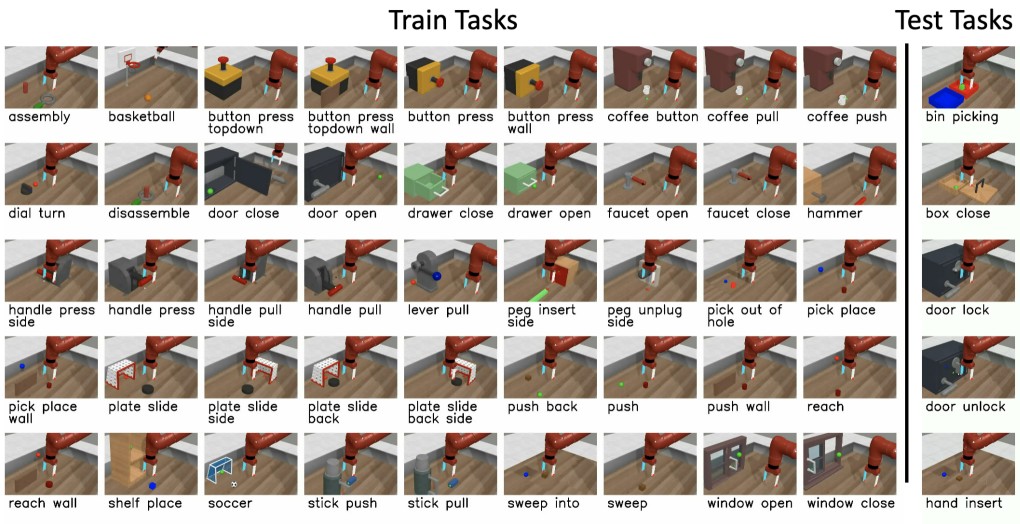

Figure 11: Meta-world Tasks in the M45 protocol with training/test Split. Figure taken from (Yu et al., 2019)

### A.5 BASELINES

**MAML PPO** is the application of the MAML meta-learning algorithm for meta-RL (Finn et al., 2017). The inner-loop algorithm is PPO, whereas the outer-loop algorithm is the MAML algorithm. We use the implementation provided in the *garage* repo (garage contributors, 2019).

**RL$^2$** uses an RNN in the inner loop as described in Section A.1, and TRPO in the outer loop (Duan et al., 2016). We use the implementation provided in the *garage* repo (garage contributors, 2019), which replaces TRPO with PPO.

**PEARL** is an off-policy meta-RL algorithm, in contrast to the other methods we compare with (Rakelly et al., 2019). We use the implementation provided in the *garage* repo (garage contributors, 2019).

**VariBAD** uses a VAE to learn a distribution of task features that increase the state space (Zintgraf et al., 2020). In contrast to PEARL, VariBAD uses PPO in the outer loop, which is an on-policy meta-RL algorithm. We use the implementation provided by the authors.

**TrMRL** (Melo, 2022) uses a transformer in the inner loop of the RL$^2$ algorithm, similar to our proposed method. However, it takes a sequence of the 5 most recent transitions as input to the transformer. We use the implementation and pipeline as provided by the authors.

**AMAGO** (Grigsby et al., 2024a) also uses a transformer, but trains it using an off-policy RL approach. It takes long sequences spanning multiple episodes as input to the transformer. We use the implementation provided by the authors.

## A.6 Additional Results

In this section, we show additional results for our evaluations on the MuJoCo, MetaWorld, and ManiSkill benchmarks.

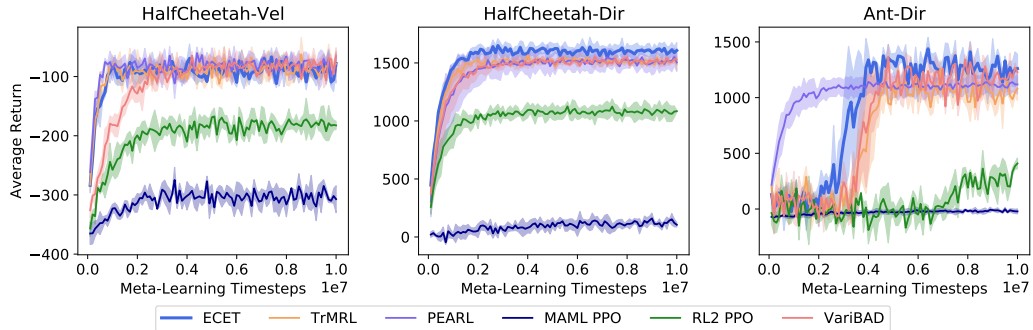

Figure 12: Meta-Train performance in terms of average return of ECET, TrMRL, PEARL, MAML PPO, RL2 PPO and VariBAD on the MuJoCo benchmark for training on parametric variations of the *HalfCheetah* and *Ant* tasks.

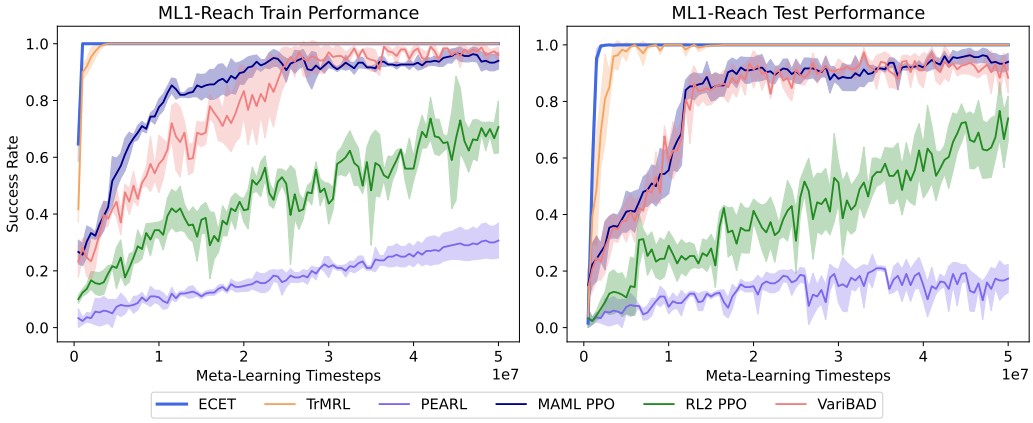

Figure 13: Meta-Train and Test performance in terms of average success rate of ECET, TrMRL, PEARL, MAML PPO, RL2 PPO and VariBAD on the ML1 benchmark for training(left) and testing(right) on parametric variations of the *Reach* task.

### A.6.1 Visualization of Embeddings

We show the visualization of the embeddings learned by ECET and TrMRL for the ML10 Training tasks in Figure 14. Furthermore, we show color maps of the inter-task distance matrix of the embeddings for TrMRL(left) and ECET(right) in Figure 15. There, it is more easily visible that the inter-task distance of the embeddings is much bigger for ECET than TrMRL, suggesting a better task differentiation.

### A.6.2 Average Ranking Plots

To focus on comparing the methods based on their robustness in relative performance to each other, in Figures 16 to 18 we provide plots showing the average ranks of the methods as a performance metric. These plots show that ECET robustly achieves the highest relative performance (and thus the lowest rank) across the meta-learning benchmarks we consider.

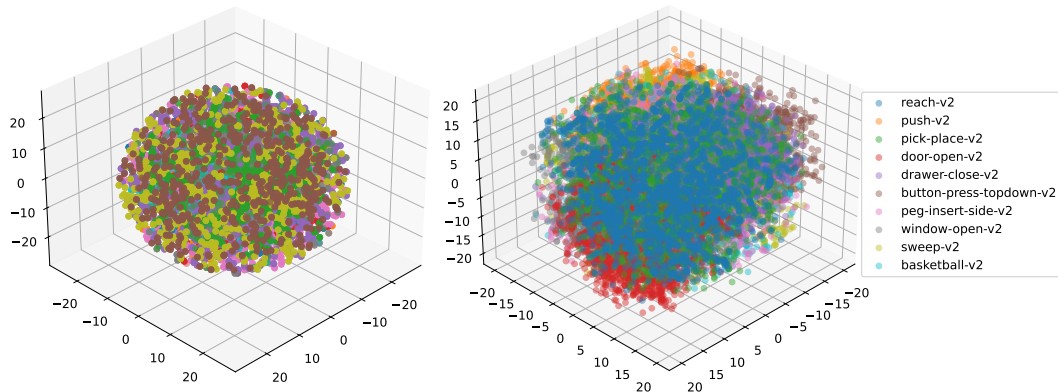

Figure 14: T-SNE plots of the output embeddings for TrMRL(left) and ECET(right) for the tasks in the training set of the ML10 benchmark of Meta-World.

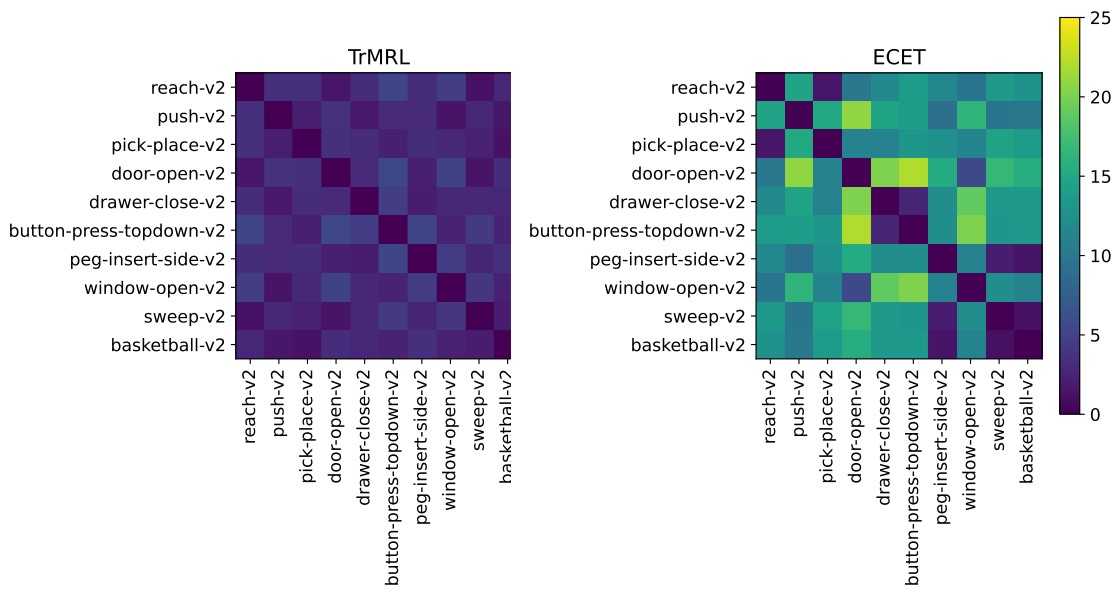

Figure 15: Color maps of the inter-task distance matrix of the embeddings for TrMRL(left) and ECET(right) for the tasks in the training set of the ML10 benchmark of Meta-World.

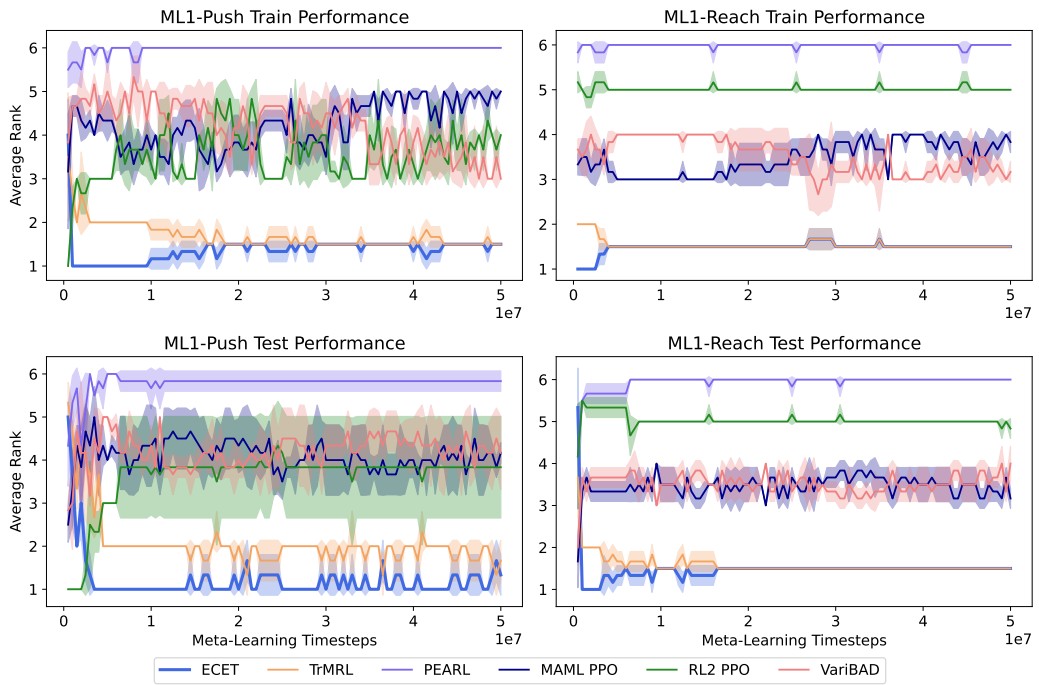

Figure 16: Average ranking plots of Meta-Train and Test performance for all methods on the ML1 *Push*(left) and *Reach*(right). Lower ranks indicate better performance.

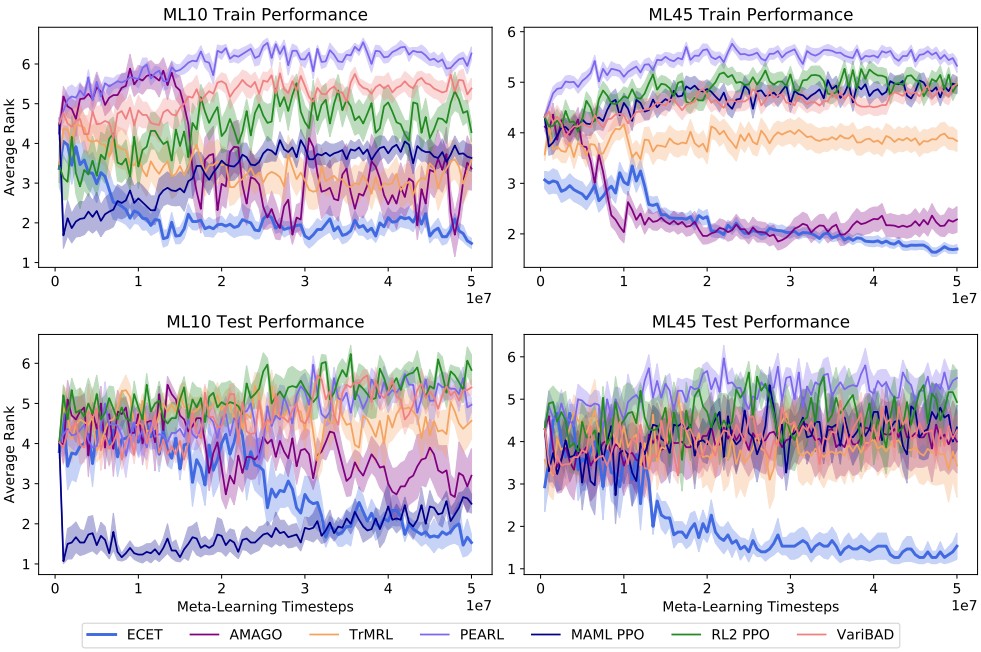

Figure 17: Average ranking plots of Meta-Train and Test performance for all methods on the ML10(left) and ML45(right) with disjoint train and test tasks. Lower ranks indicate better performance. The corresponding success rate plot can be found in Figure 6.

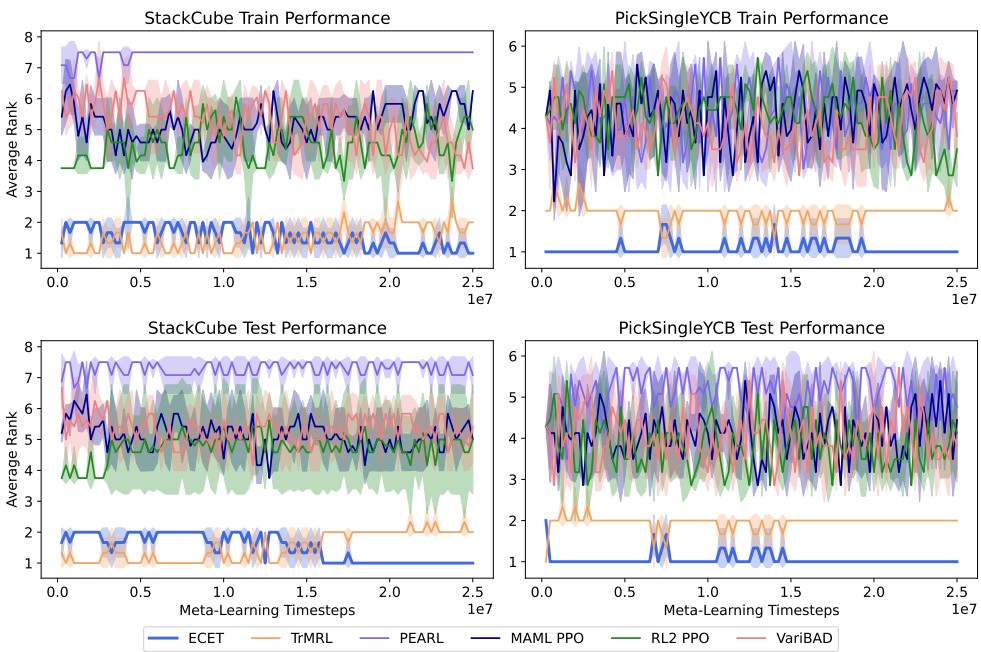

Figure 18: Average ranking plots of Meta-Train and Test performance for all methods on the StackCube(left) and PickSingleYCB(right) tasks with disjoint train and test tasks. Lower ranks indicate better performance.

### A.6.3 PERFORMANCE WITH RESPECT TO TIME

We show performance of the methods in terms of average success rate with respect to time in GPU hours in Figures 19 to 22. These plots show that in most cases ECET achieves the best anytime performance across the meta-learning benchmarks we consider.

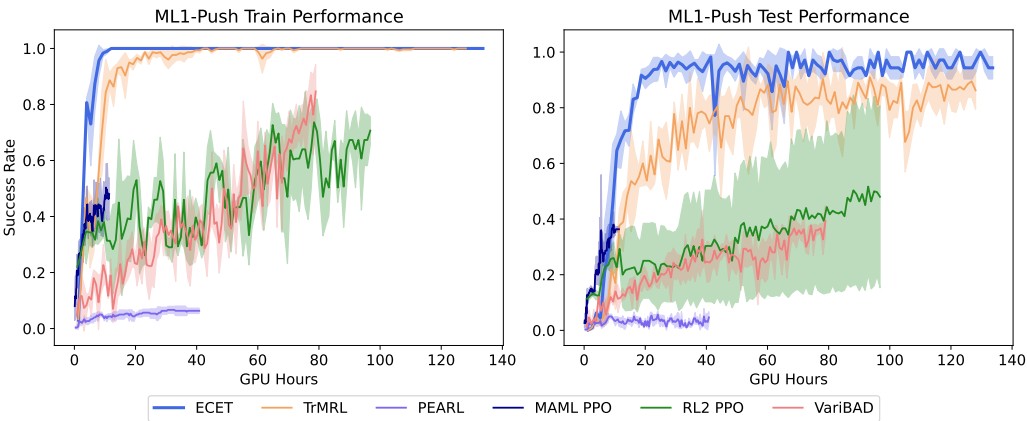

Figure 19: Meta-Train and Test performance in terms of average success rate with respect to time in GPU hours for all methods on the ML1 *Push* task.

### A.6.4 EVALUATION ACROSS EPISODES IN TEST TASKS

To assess the test performance of ECET, we perform rollouts for 50 episodes on test tasks and compare the episodic return for each episode to the baselines.

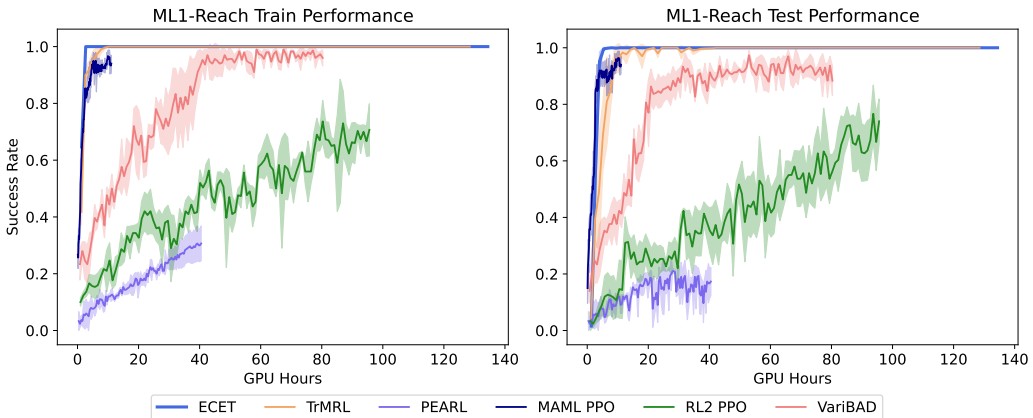

Figure 20: Meta-Train and Test performance in terms of average success rate with respect to time in GPU hours for all methods on the ML1 *Reach* task.

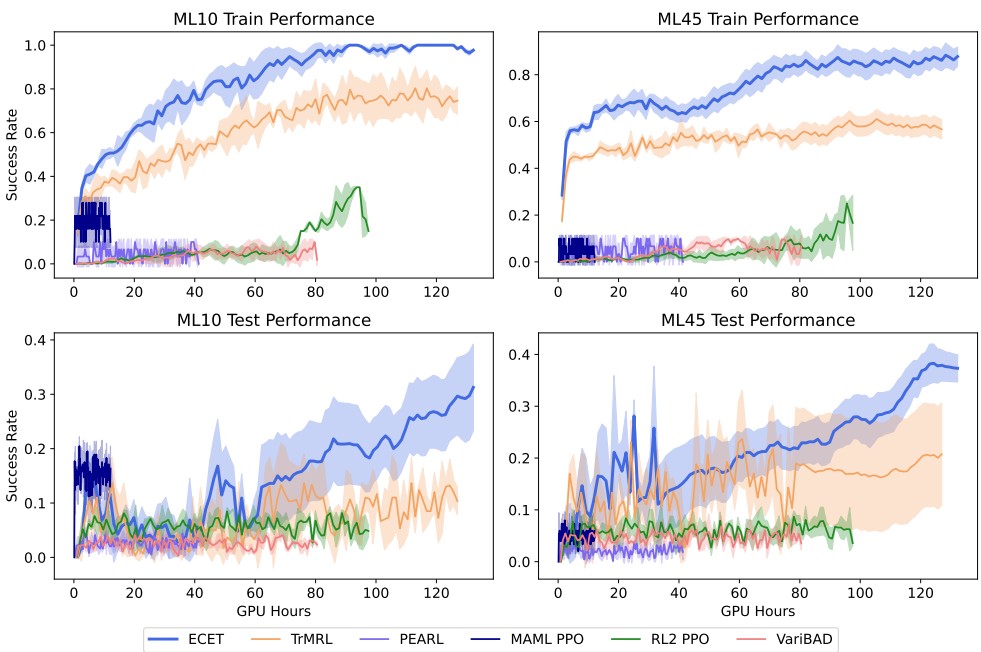

Figure 21: Meta-Train and Test performance in terms of average success rate with respect to time in GPU hours for all methods on the ML10(left) and ML45(right) with disjoint train and test tasks.

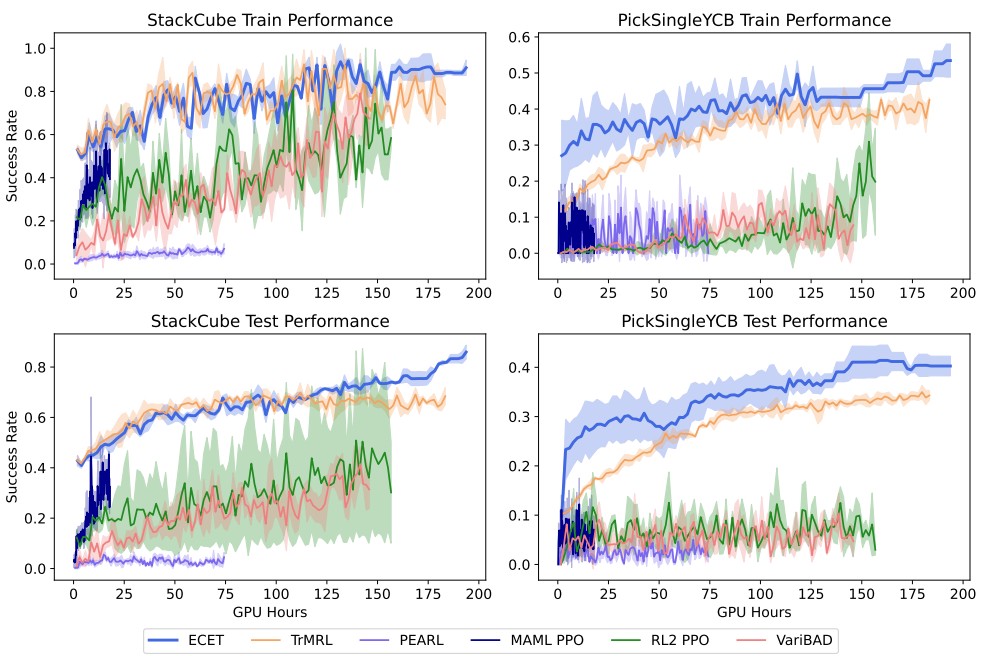

Figure 22: Meta-Train and Test performance in terms of average success rate with respect to time in GPU hours for all methods on the StackCube(left) and PickSingleYCB(right) tasks with disjoint train and test tasks.

**ML1** We sample 10 test tasks for *Push-v2* and perform rollouts of 50 episodes with each of the resulting models from the meta-training using 5 seeds for each method. For each episode we plot the mean and standard error. Figure 23 shows the resulting plots, where it is visible that ECET generalizes well accross parametric variations it has not seen in meta-training, by achieving the highest episodic returns consistently accross episodes. This consistency since the first episode shows the independence of ECET to the experience available for the current task. This happens because during meta-training, we make sure to train on transitions where we don't have the full history of 25 episodes available.

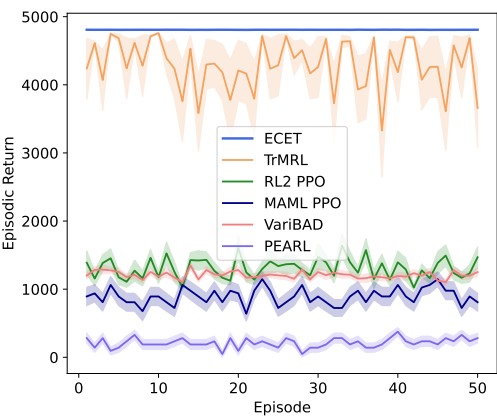

Figure 23: Episodic return accross episodes on test tasks for all methods on the ML1 *Push* Benchmark.

**ML10** We sample 10 parametric variations for each of the tasks in the test set of ML10 (*Door Close*, *Sweep Into*, *Lever Pull*, *Shelf Place*, *Drawer Open*) and perform rollouts of 50 episodes with each of the resulting models from the meta-training using 5 seeds for each method. For each episode

we plot the mean and standard error. Figure 24 shows the resulting plots. It is visible that ECET generalizes well accross tasks it has not seen in meta-training, by achieving the highest episodic returns consistently accross episodes. However, for some tasks, there is more variance in performance accross seeds, as well as accross episodes, which indicates more difficulty to adapt. Similarly as for the ML1 benchmark, there is no drop in performance for ECET in the first episodes where we have less observations, further demonstrating the independence of ECET to the experience available for the current task.

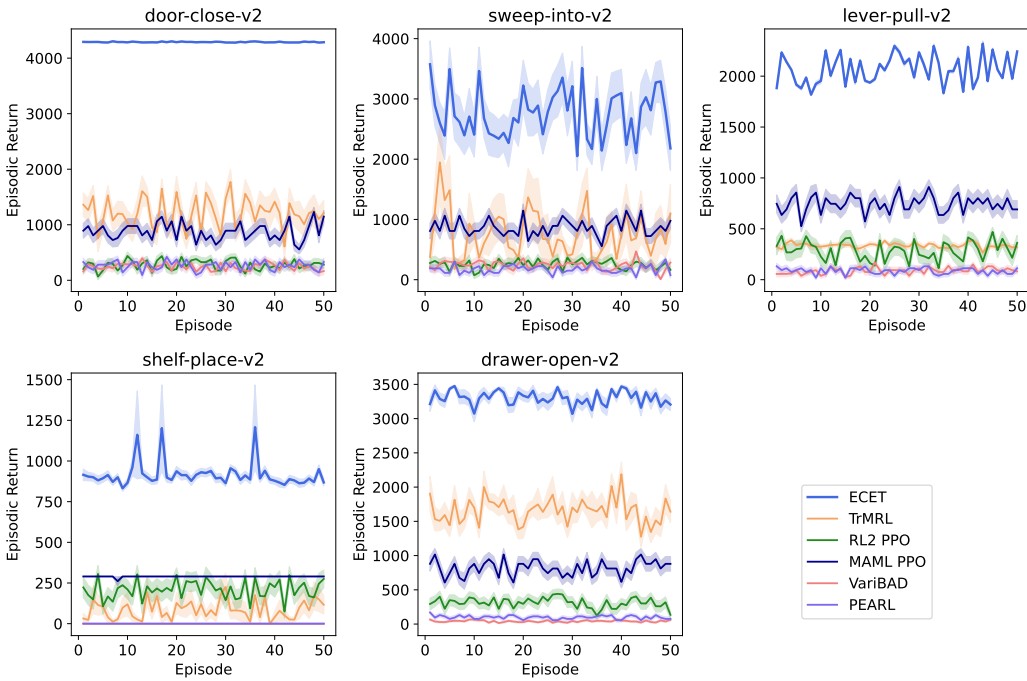

Figure 24: Episodic return accross episodes on each test task for all methods on the ML10 Benchmark.

**ML45** We sample 10 parametric variations for each of the tasks in the test set of ML10 (*Box Close*, *Bin Picking*, *Door Unlock*, *Hand Insert*, *Door Lock*) and perform rollouts of 50 episodes with each of the resulting models from the meta-training using 5 seeds for each method. For each episode we plot the mean and standard error. Figure 25 shows the resulting plots. Compared to the baselines, ECET manages to get higher episodic returns on tasks it has not seen in meta-training, similarly to the ML1 and ML10 benchmark. However, for ML45, the variance in performance is higher accross episodes, which indicates more difficulty to adapt. For the *Door Unlock* task, it is noticeable that TrMRL manages to perform similarly to ECET. The other baselines fail to reach high episodic return. Similarly as for the ML1 benchmark, there is no drop in performance for ECET in the first episodes where we have less observations, further demonstrating the independence of ECET to the experience available for the current task.

### A.6.5 ABLATIONS

To investigate the behavior of ECET and the impact of its components on performance, we ablate the number of episodes $E$ to keep in memory and the number of transitions $T$ in the sampled sequence in Figure 28. The configuration that performs best, and the one we use in our experiments is $E = 25$, $T = 5$. We ablate the sampling strategy (sampling one transition sequence per episode vs. sampling multiple sequences randomly from any episode) in the top plot in Figure 29, to show the usefulness of sampling one transition sequence per episode. In contrast to TrMRL, which only uses the output of the transformer as input to the policy $\pi$, we concatenate a linear transformation of the current state as input to the policy. We ablate the performance of ECET for these two settings of policy input in the middle plot in Figure 29. In the bottom plot, we ablate the number of encoder blocks for IET and CET. To differentiate the contributions of the context sampling strategy from the architecture in

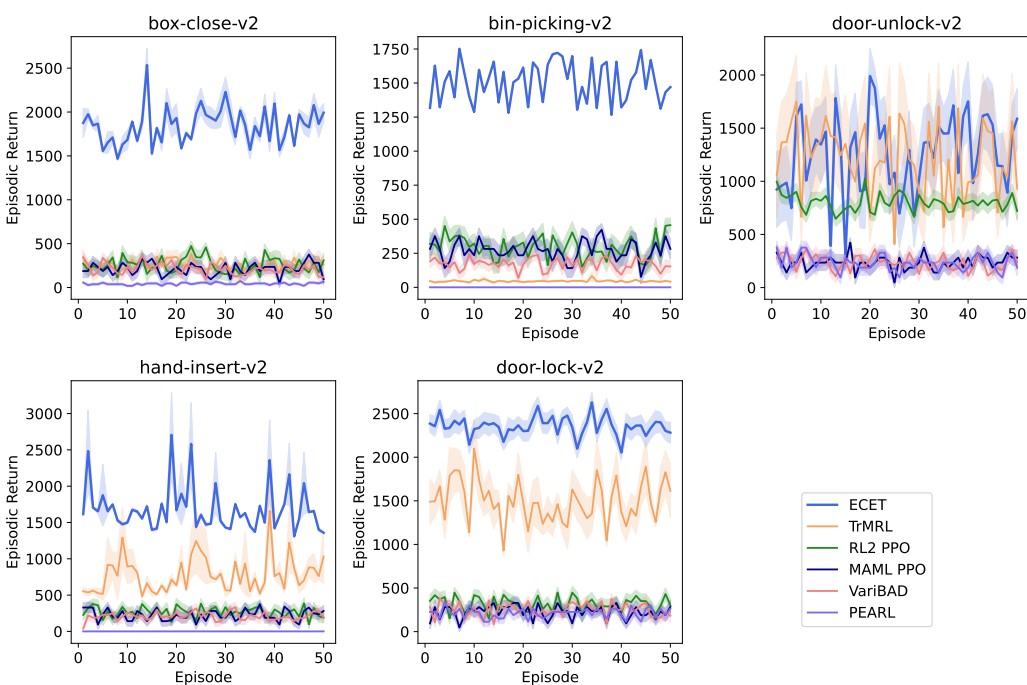

Figure 25: Episodic return accross episodes on each test task for all methods on the ML45 Benchmark.

ECET, in Figure 26 we compare ECET to a variant using a positional encoding for CET (ECET (pos)) and one using a flat architecture with only IET (ECET (flat)). This version processes context that is sampled exactly the same as in ECET, specifically sequences of length S are randomly sampled from E episodes and then put through the transformer. ECET (flat) yields inferior performance, underscoring the challenges a non-hierarchical approach faces in capturing comprehensive task representations. Finally, in Figure 27 we also compare ECET to TrMRL, and TrMRL (extended context) that processes sequences of the same length as the total number of transitions that ECET processes, namely sequences of length ExS. We extend the context of TrMRL to investigate the effect that the context length has on the performance improvements we see with ECET. The low success rate of TrMRL (extended context) shows that extending the context length of TrMRL makes the meta-training less efficient, as much more training data is needed to extract useful task meta-features from these long sequences. Our benchmark configuration for these experiments stands at $E = 25$, $T = 5$,incorporates two encoder blocks for both **IET**(with positional encoding) and **CET**(without positional encoding), and uses a linear transformation $\varphi_2$ of the state as additional input to the policy $\pi$. We do all our ablations in the ML1 Benchmark, using parametric variations of the *Push* task.

Figure 26 shows the performance of our default configuration of the method, ECET, compared to using a positional encoding for **CET**, ECET (pos), and the simplification to a flat architecture akin to TrMRL by solely employing **IET**, ECET (flat). We show the average and standard error of the success rate accross Meta-Learning Timesteps. ECET (flat) performs worst, indicating the difficulty of a flat architecture to capture a better representation of the task, instead of focusing on local charactersitics of the dynamics. ECET (pos) has a similar rate of improvement to ECET in the beginning, but ends up converging to a suboptimal representation of the task, and thus its success rate does not improve above $\sim 0.8$ after $2.5 \cdot 10^6 Timesteps$. This indicates that the postional encoding in **CET** induces an ordering in the $\mathbf{z}_{seq}$ representations which is misleading, as episodes don't naturally have a causal ordering, thus imposing an ordering on the representations we get from the sequences of transitions sampled ends up hurting performance.

Figure 27 shows the performance of our default configuration of the method, ECET, compared to TrMRL that processes sequences of length $5$, and TrMRL (extended context) that processes sequences of length $125$. We extend the context of TrMRL to investigate the effect that the context length has on the performance improvements we see with ECET. The low success rate of TrMRL (extended

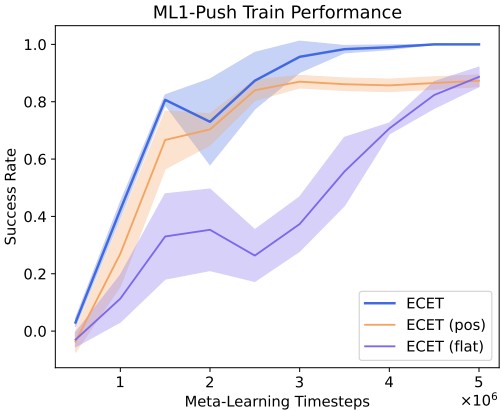

Figure 26: Investigating the meta-training performance impact on ECET of the inter-episode feature extractor **CET** and the positional encoding on the sequence embeddings $\mathbf{z}_{seq}$ before being input into **CET** in the ML1 *Push* Benchmark. ECET shows the performance of the configuration we propose, with **IET** using positional encoding, and **CET** without positional encoding. ECET (pos) shows the performance of the configuration with **IET** and **CET** both using positional encoding. ECET (flat) shows the performance of the configuration with no hierarchy of transformers, only **IET** using positional encoding, taking the same information as input as ECET.

context) shows that extending the context length of TrMRL makes the meta-training less efficient, as much more training is needed to extract useful task meta-features from these long sequences. Thus, the ECET architecture of encoding shorter sequences from different episodes with IET, and then processing those encodings with CET is crucial in ensuring better anytime performance than TrMRL.

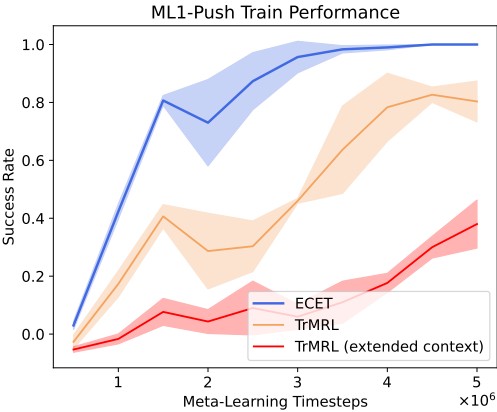

Figure 27: Performance of ECET, TrMRL with its standard sequence length of 5, and TrMRL with extended context (processing sequences of length 125 to equal the context length of ECET).

To investigate the performance sensitivity of ECET on the number of episodes (E) kept in memory and the samples sequence length (T), we provide ablation results in Table 2.

Figure 28 shows the performance of ECET, when varying key parameters and structural elements: the number of episodes ($E$) in memory, the length of sampled transition sequences ($T$). The top plot compares the impact of varying $E = \{5, 10, 15, 20, 25, 50\}$ when $T = 5$. It is noticeable that increasing $E$ to 50 results in suboptimal performance, as the configuration does not manage to reach a success rate of 1 compared to the other configurations. On the other side of the spectrum, decreasing $E$ to 5 results in a slower improvement. The configuration that we use, $E = 25$, shows

| T \ E | 5 | 10 | 15 | 20 | 25 | 50 |
|---|---|---|---|---|---|---|
| 5 | $0.997 \pm 0.005$ | $1.000 \pm 0.000$ | $0.997 \pm 0.005$ | $1.000 \pm 0.000$ | $1.000 \pm 0.000$ | $0.909 \pm 0.058$ |
| 25 | $0.997 \pm 0.005$ | $1.000 \pm 0.000$ | $1.000 \pm 0.000$ | $0.993 \pm 0.009$ | $0.924 \pm 0.053$ | $1.000 \pm 0.000$ |
| 50 | $0.963 \pm 0.039$ | $0.960 \pm 0.050$ | $0.993 \pm 0.005$ | $0.827 \pm 0.098$ | $0.990 \pm 0.014$ | $0.893 \pm 0.041$ |

Table 2: ECETs final meta-training success-rate on **ML1 Push** when varying the number of episodes (E) and the sequence length (T).

the fastest improvement and better performance accross the Meta-Learning Timesteps. The middle plot compares the impact of varying $E = \{5, 10, 15, 20, 25, 50\}$ when $T = 25$. It is noticeable that all of the possible configurations are slower to improve than the configurations using $T = 5$, and that is because with higher sequence length, ECET needs much more data and training to arrive at a good representation $\mathbf{z}_{seq}$ and $\mathbf{z}_{task}$. This claim is further supported by the bottom figure, showing even slower improvement for configurations with $E = \{5, 10, 15, 20, 25, 50\}$ when $T = 50$.

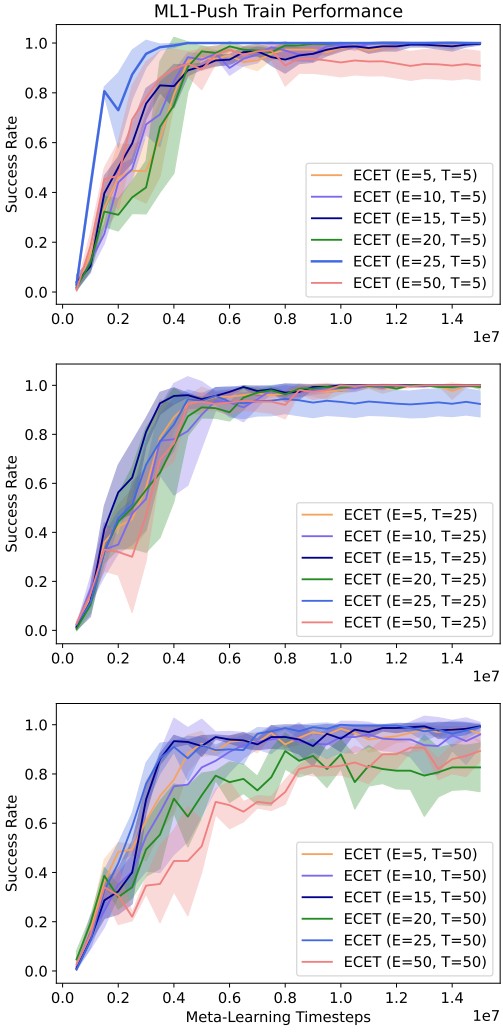

Figure 28: Investigating the meta-training performance impact of the sequence length T and number of episodes E for ECET in ML1 *Push*. The top plot shows the performance of sequence lengths $T = 5$ with different episode lengths $E \in \{5, 10, 15, 20, 25, 50\}$. The middle plot shows the performance of $S = 25$ with $E \in \{5, 10, 15, 20, 25, 50\}$. For the bottom plot, $T = 50$.

Figure 29 top shows ablations of our sampling strategy—selecting a singular transition sequence per episode to a more randomized approach drawing multiple sequences from any episode. All of the shown configurations use the latter strategy, as in Figure 28 and everywhere else we use the former strategy as our default. It is noticeable that when sampling sequences from any randomly chosen episode, the performance improvement is slower. However, for the $E = 50, T = 5$ configuration, the randomization of sampling helps ECET not get stuck in suboptimal regions, and thus it manages to reach a success rate of $1.0$ compared to the same configuration using the non-randomized strategy of sampling with respect to episodes, which we see in Figure 28. Despite this, all the configurations using the randomized strategy have slower improvement compared to our default configuration using the non-randomized strategy. The middle plot shows the performance comparison of augmenting the policy $\pi$ input with a linear transformation of the state(ECET) versus relying exclusively on the transformer's output(ECET*). We notice a significant improvement from getting the linear transformation of the state as input in the policy. The bottom plot shows the influence of varying the number of encoder blocks within the architecture. Although it has least parameters compared to the other configurations, ECET(2, 2) performs best, having the fastest rate of improvement and also best final performance.

### A.6.6 RANK AND SUCCESS RATE PLOTS PER TASK

Through investigating the individual plots for each task in terms of average success rate in Figures 30-34, the difference in the level of difficulty for the tasks can be demonstrated, noticing that e.g. in Figure 31 the *Door Close* task is easier to successfully perform after meta-training, reaching success rates of approximately $0.6$, compared to the *Shelf Place* task, for which all the methods fail to reach a success rate higher than $0.12$. This is also the case for the performance per task for the test tasks in ML45, which we show in Figure 34. For the *Bin Picking* task, none of the baselines can successfully finish even one episode, except for VariBAD with success rate of approximately $0.05$ (ECET being more robust, reaching $0.3$). On the other hand, for the *Door Unlock* task, RL2 PPO manages to reach success rates of approximately $0.4$, and TrMRL reaches $0.6$, being one of the few cases where ECET underperforms by reaching a success rate of $0.3$.

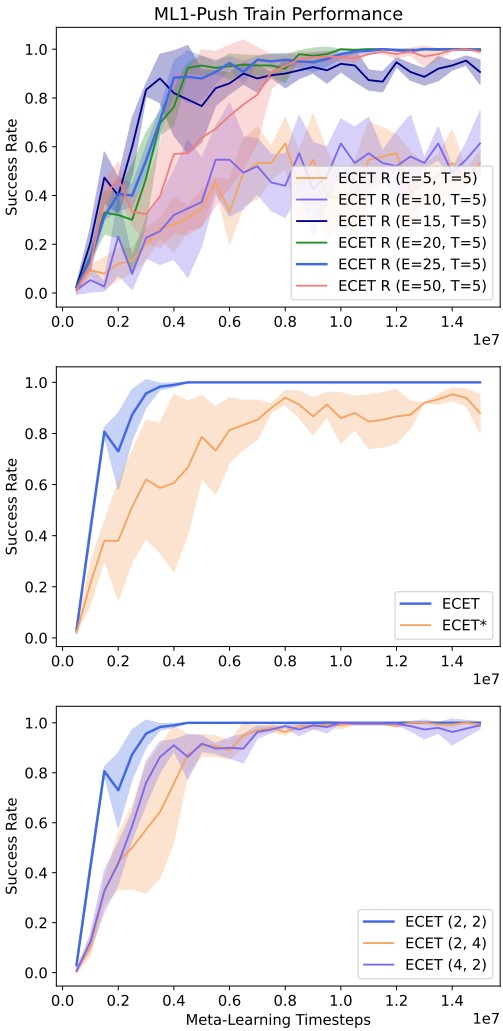

Figure 29: Investigating the meta-training performance impact of the sampling strategy for the sequence for ECET in ML1 *Push*. The top plot shows the performance of ECET when sampling $E$ sequences from random episodes (i.e. more than one sequence per episode). The middle plot shows the performance impact of concatenating a linear transformation of the state(ECET) vs using only the output of the transformer(ECET*) as the input for the policy. The bottom plot shows the performance impact of the number of encoder blocks, comparing configurations of 2 encoder blocks for each **IET** and **CET**, 2 encoder blocks for **IET** and 4 for **CET**, and 4 encoder blocks for **IET** and 2 for **CET**.

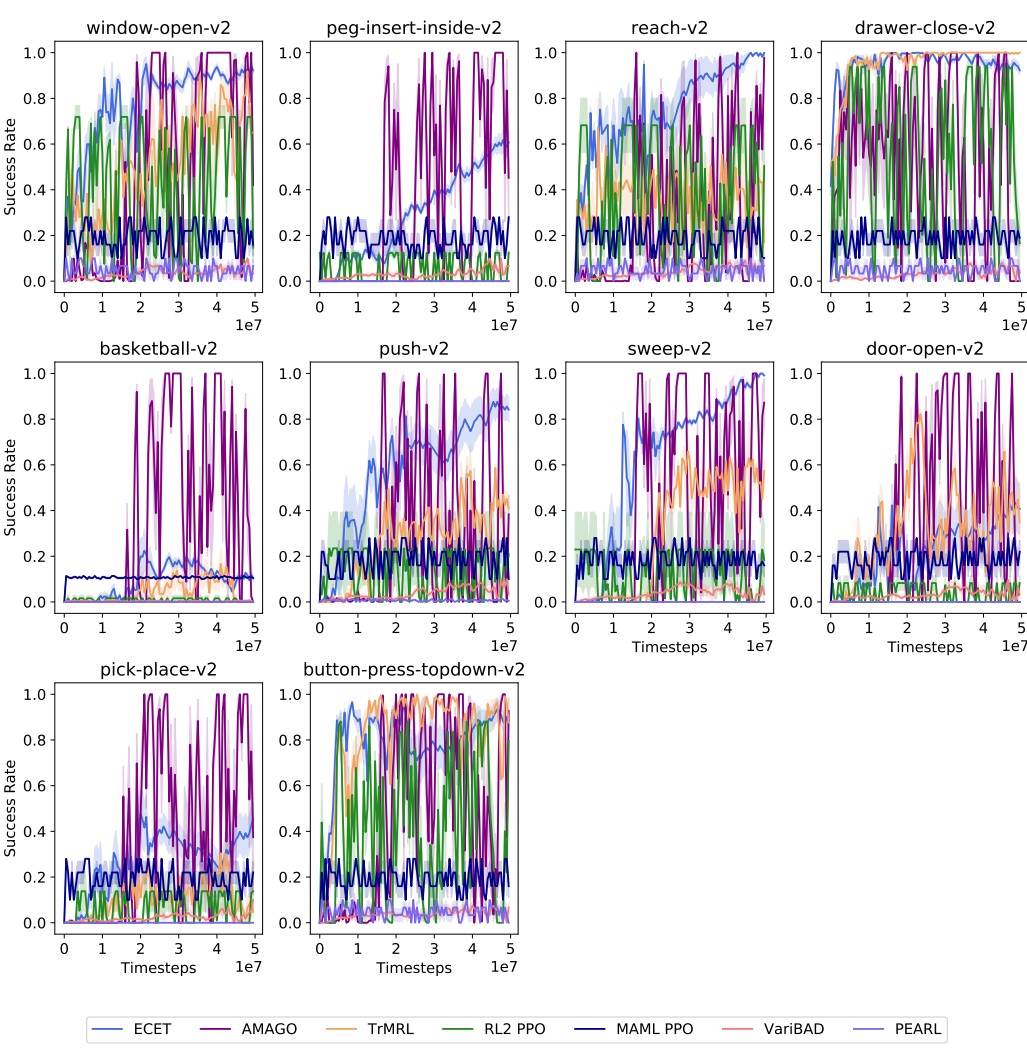

Figure 30: Success rate per train task for the ML10 benchmark for ECET, TrMRL, PEARL, RL2 PPO, MAML PPO, and VariBAD.

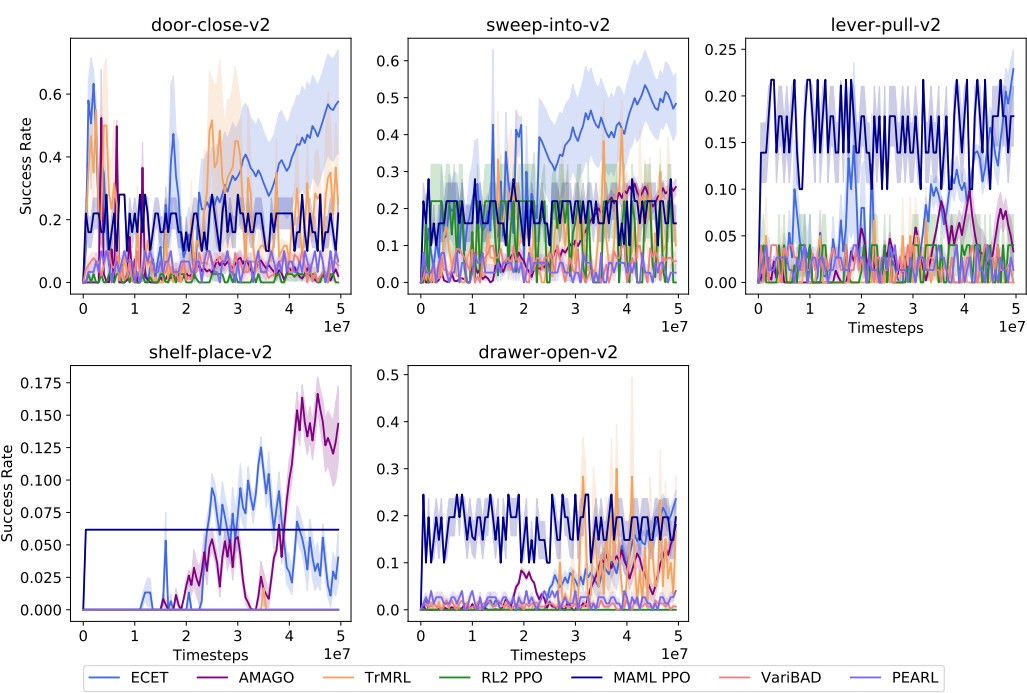

Figure 31: Success rate per test task for the ML10 benchmark for ECET, TrMRL, PEARL, RL2 PPO, MAML PPO, and VariBAD.

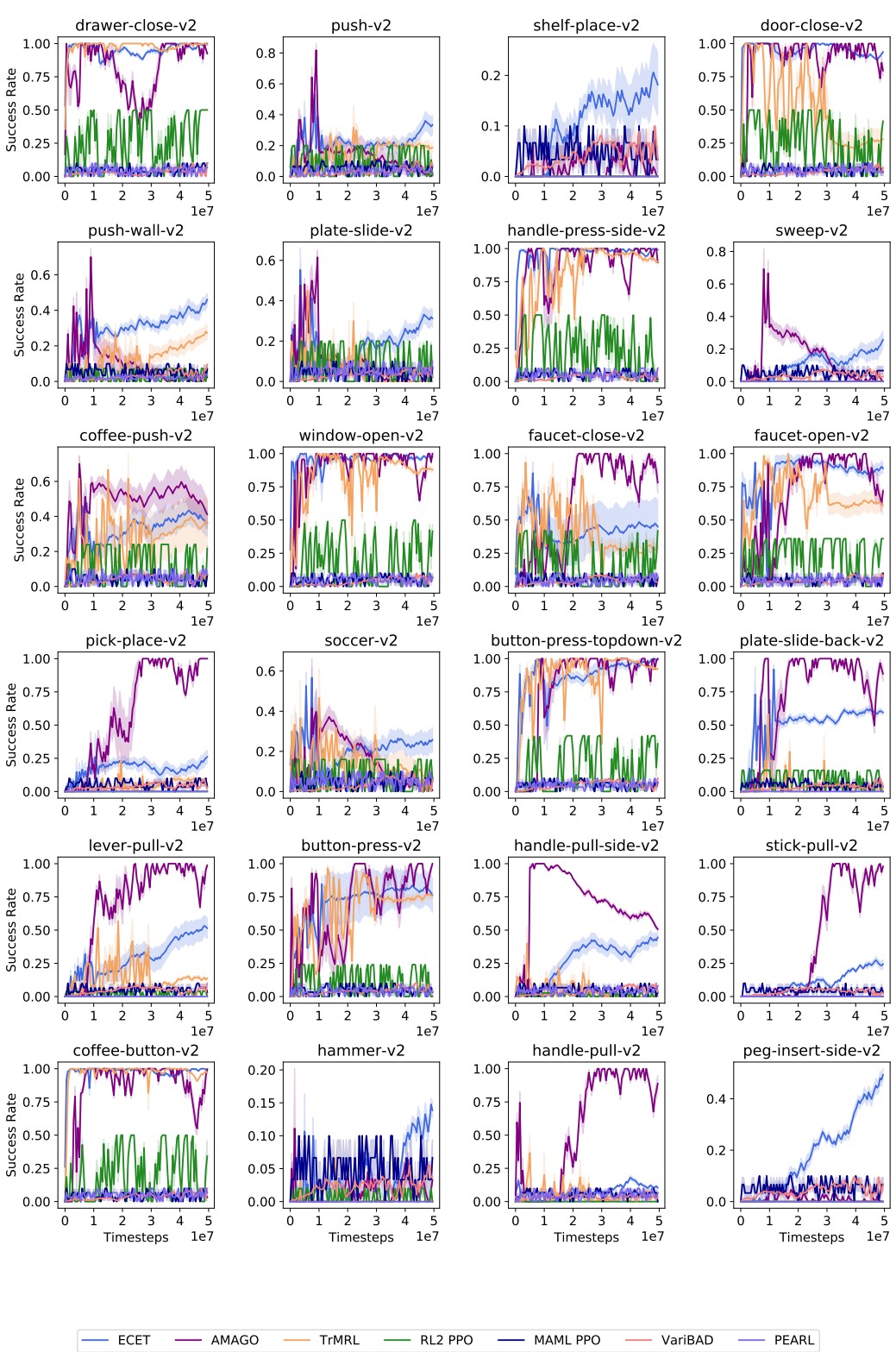

Figure 32: Success rate per train task for the ML45 benchmark for ECET, TrMRL, PEARL, RL2 PPO, MAML PPO, and VariBAD (Part 1).

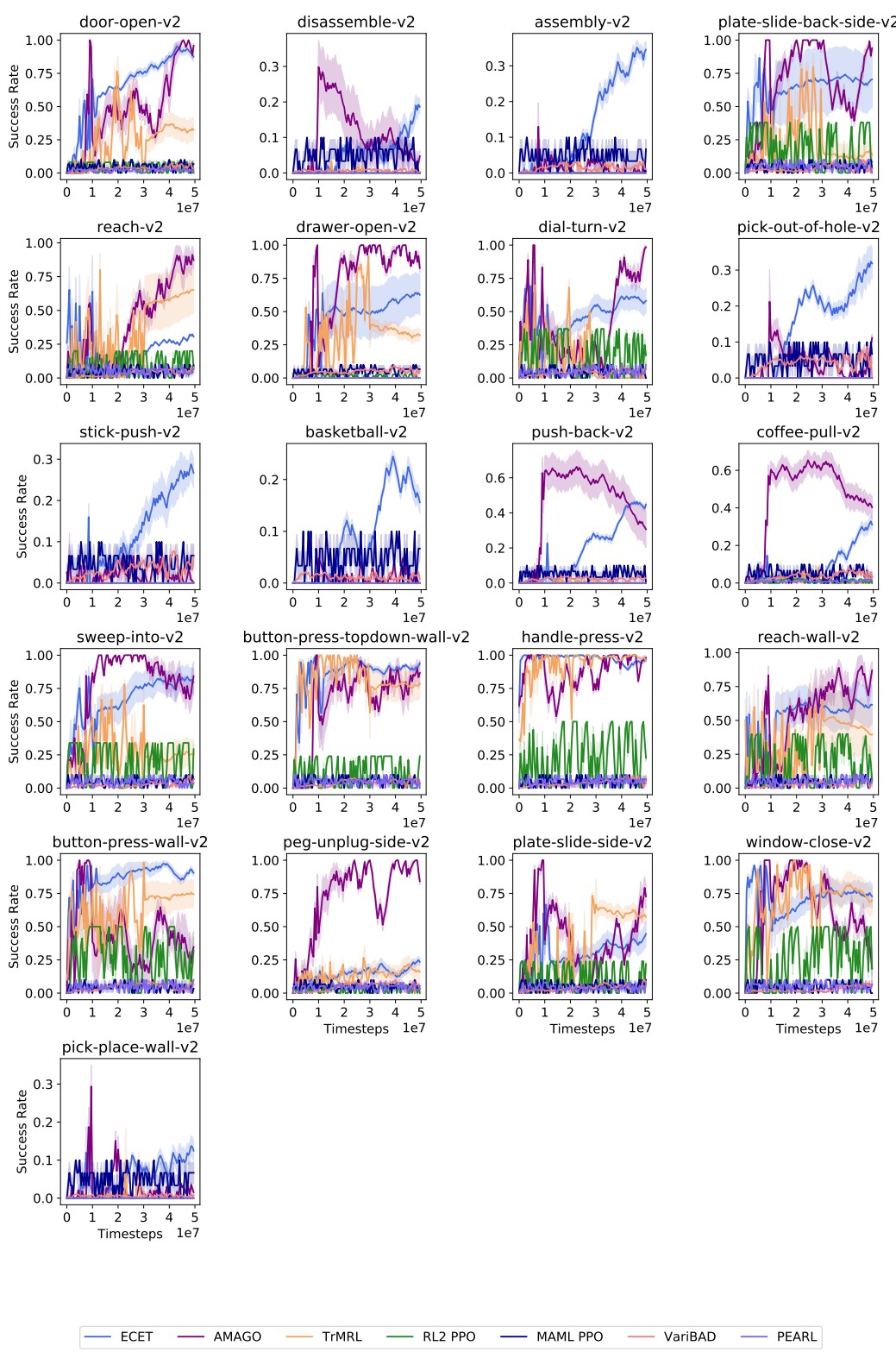

Figure 33: Success rate per train task for the ML45 benchmark for ECET, TrMRL, PEARL, RL2 PPO, MAML PPO, and VariBAD (Part 2).

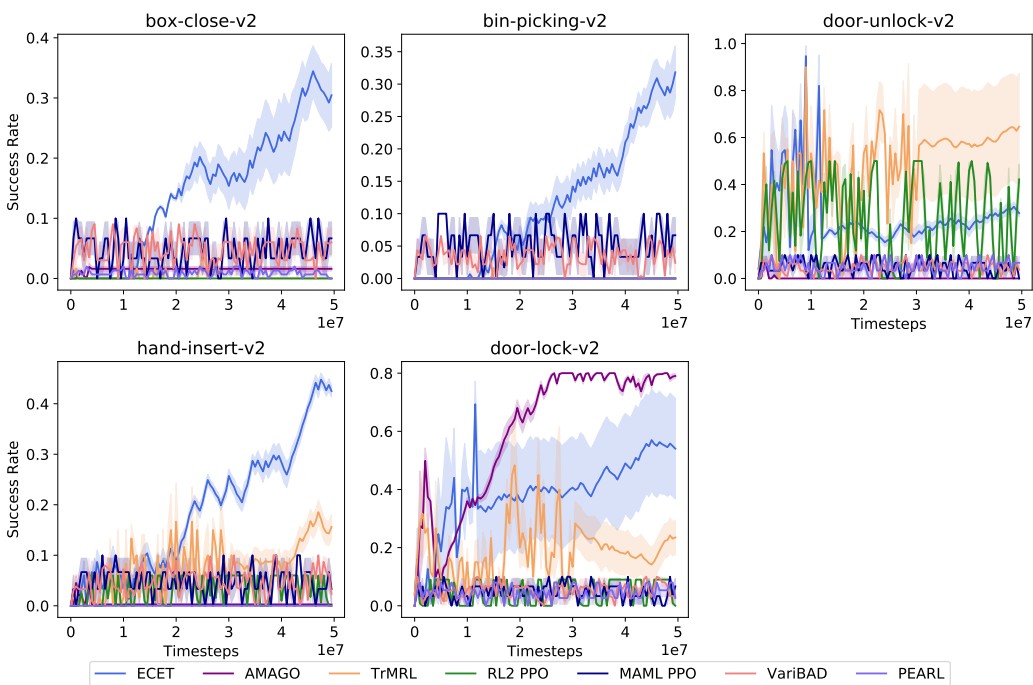

Figure 34: Success rate per test task for the ML45 benchmark for ECET, TrMRL, PEARL, RL2 PPO, MAML PPO, and VariBAD.

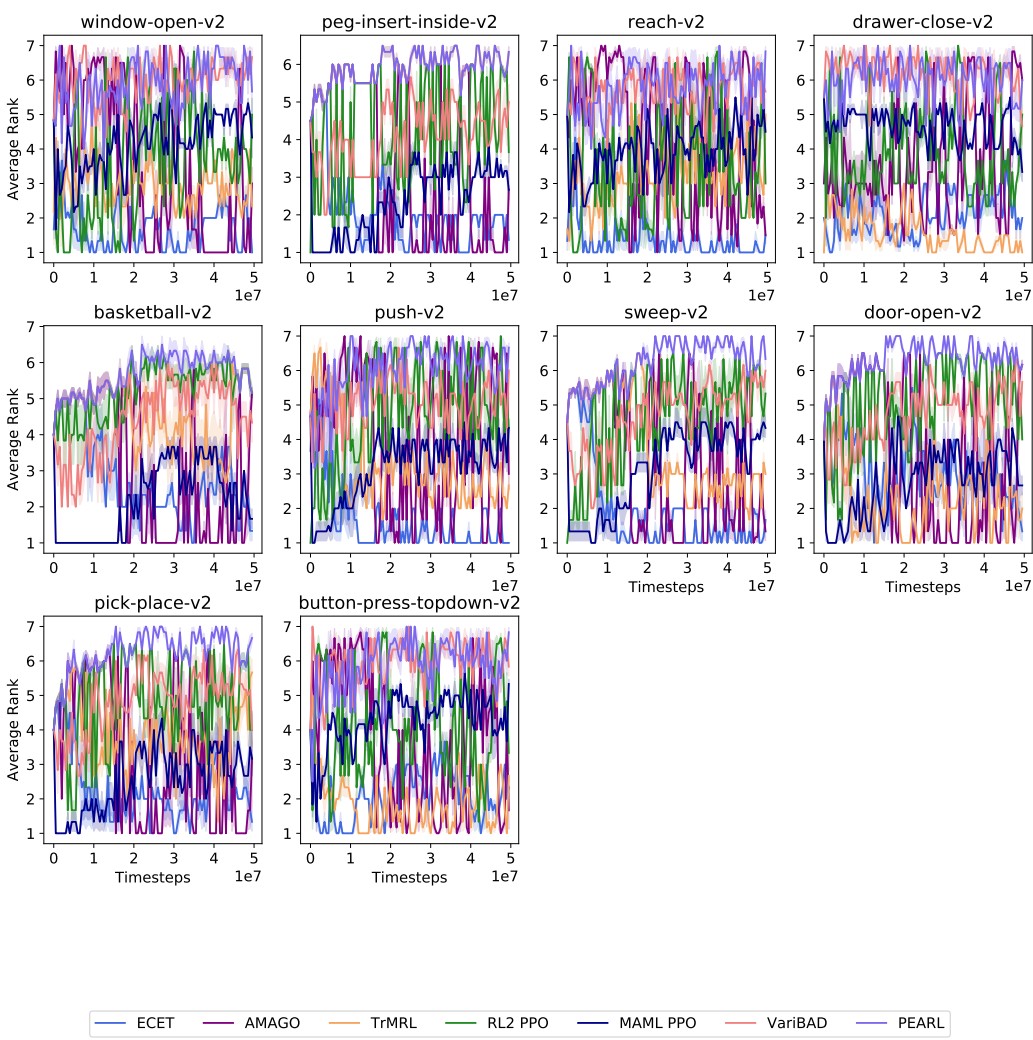

Figure 35: Average rank per train task for the ML10 benchmark for ECET, TrMRL, PEARL, RL2 PPO, MAML PPO, and VariBAD.

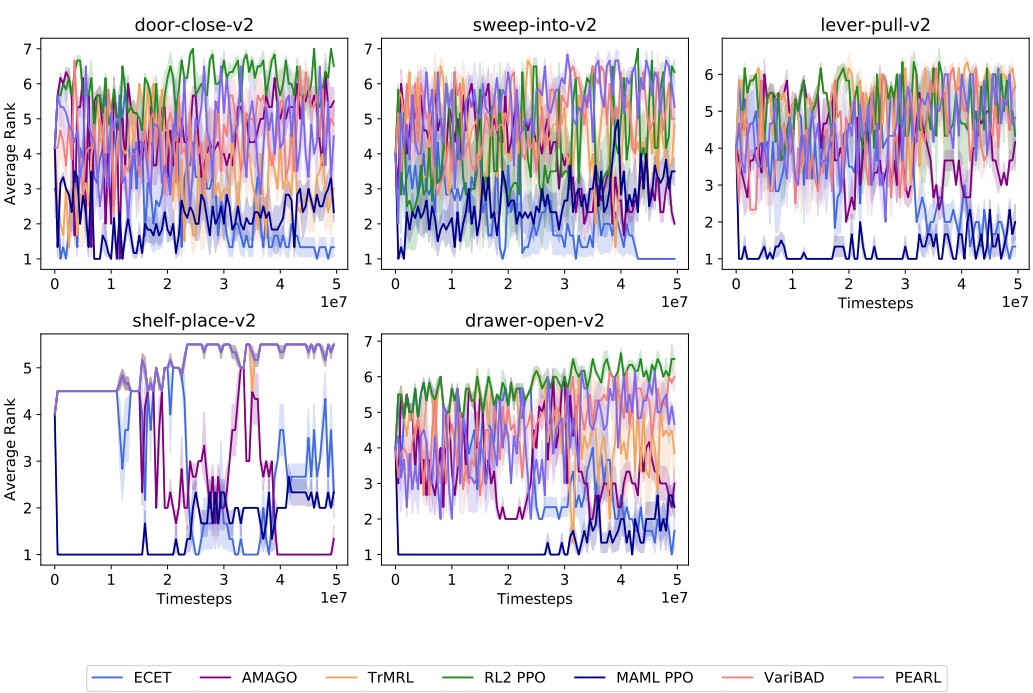

Figure 36: Average rank per test task for the ML10 benchmark for ECET, TrMRL, PEARL, RL2 PPO, MAML PPO, and VariBAD.

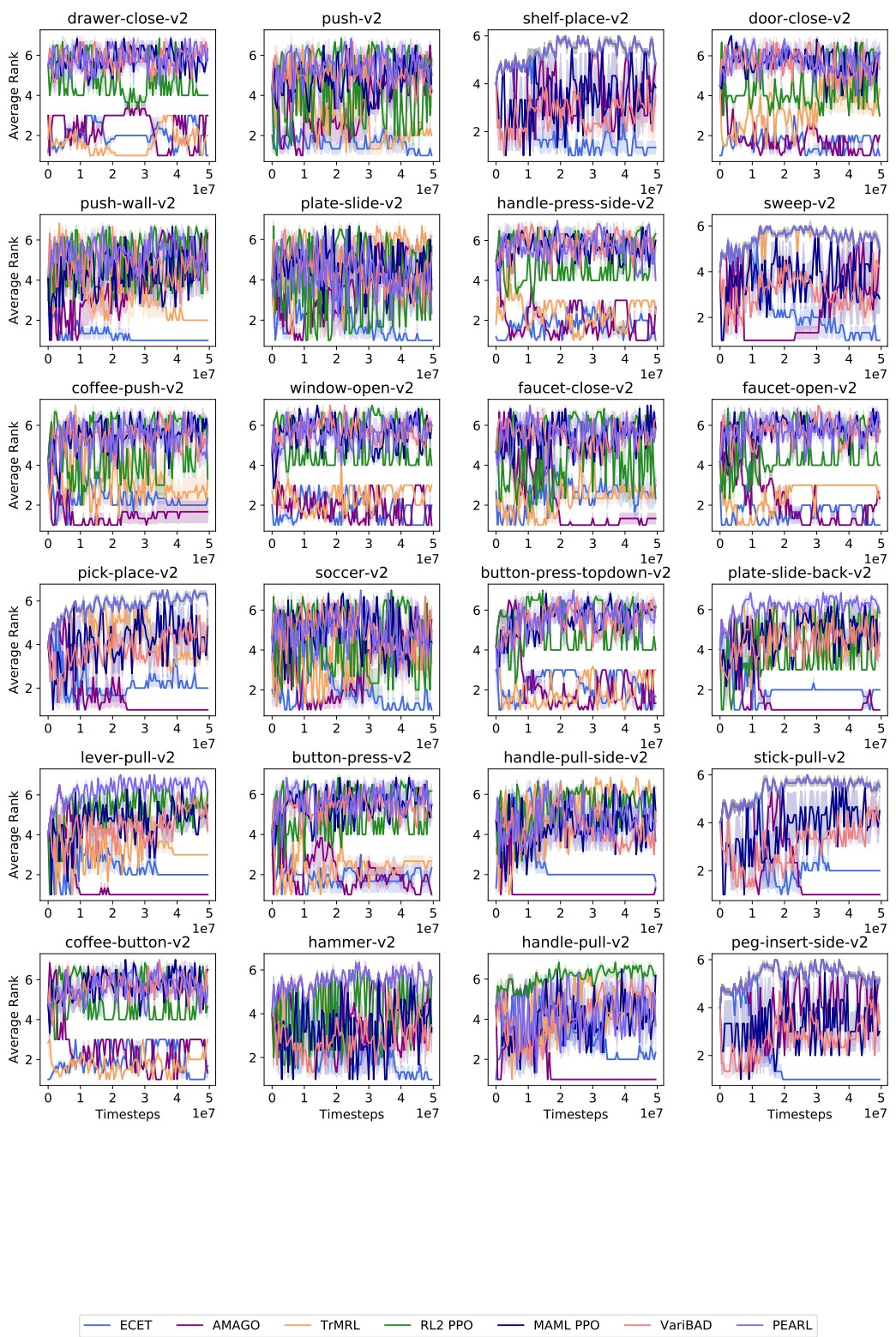

Figure 37: Average rank per train task for the ML45 benchmark for ECET, TrMRL, PEARL, RL2 PPO, MAML PPO, and VariBAD (Part 1).

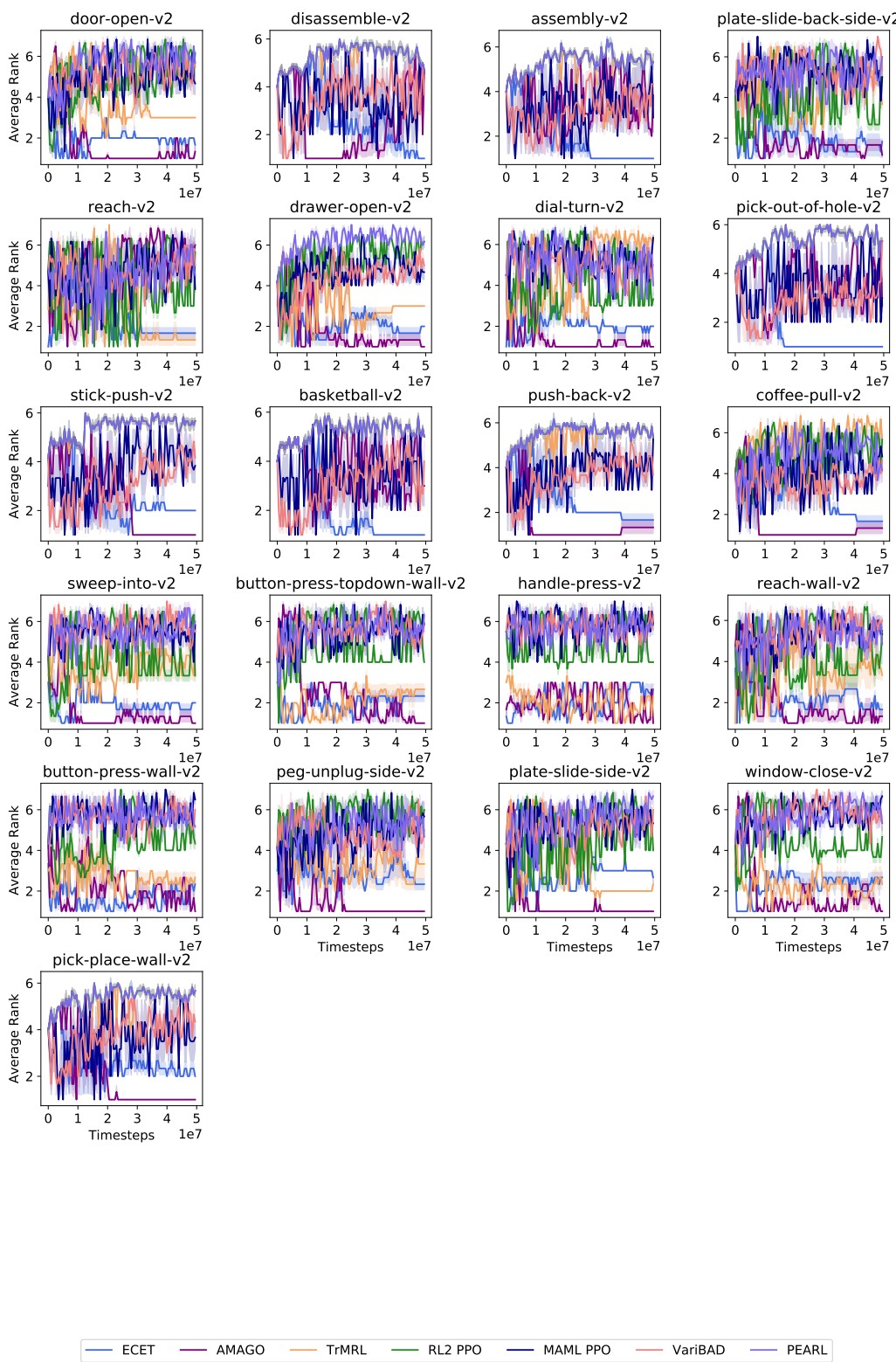

Figure 38: Average rank per train task for the ML45 benchmark for ECET, TrMRL, PEARL, RL2 PPO, MAML PPO, and VariBAD (Part 2).

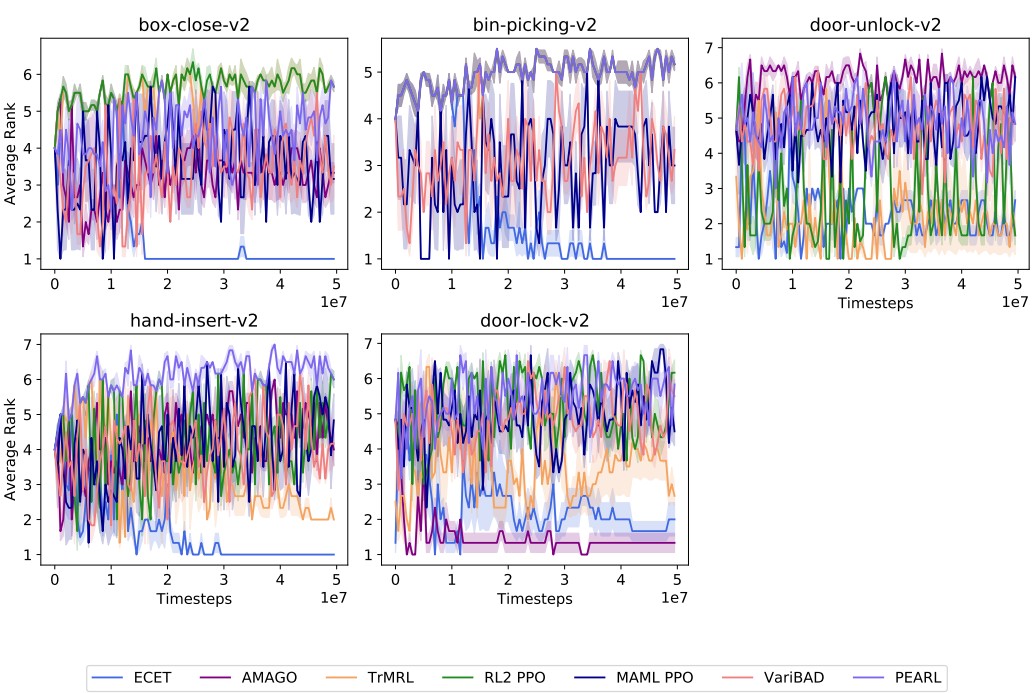

Figure 39: Average rank per test task for the ML45 benchmark for ECET, TrMRL, PEARL, RL2 PPO, MAML PPO, and VariBAD.

