# OpenReview forum: "Efficient Cross-Episode Meta-RL"
_ICLR.cc/2025/Conference — ICLR 2025 Poster_

### Official Review · Reviewer_URVA · 2024-10-25

**Soundness:** 2
**Presentation:** 2
**Contribution:** 1
**Rating:** 3
**Confidence:** 5

**Summary:**

This paper presents an online Meta-RL method. The authors suggested that decomposing the encoder into an intra-episode stage (where each episode is encoded separately) and cross-episode stage (where the vectors from the previous step are encoded into a single representation of the context) should improve performance both in terms of few-shot adaptation and in-distribution and out-of-distribution generalization.
The authors validated their claim on the Meta-World and ManiSkill benchmarks and compared against popular Meta-RL benchmarks.

**Strengths:**

1. The paper is well written, especially the Method and Experiments sections. The formulation and explanation of the algorithm can be well understood
2. The results shown in the Experiments section look quite promising, including results over the widely used Meta-World benchmark
3. The authors provided an open-sourced code to reproduce the results

**Weaknesses:**

1) My main concern regarding this work is the novelty of the proposed method. The overall algorithm is very similar to previous lines of work, especially RL2 and TrMRL with the main difference from TrMRL is the decomposition of the transformer into two stages. For this change to have a profound enough impact on the field I would expect at least one of the following:

      a) A theoretical/analytical analysis of why this specific architecture should generalize/perform better than others. The authors gave high-level explanations (e.g lines 204-209), but I found these not rigorous enough. For example, there is no sufficient motivation for hypothesis 1 (line 368), regular recurrent architectures should also be able to capture intra and cross episodic experiences, why should we expect ECET do it better? \
      b) A more thorough empirical study showing the benefits of the chosen design, including more baselines, benchmarks, and an ablation study (more on that below)

2) The positioning of the paper concerning related work is not clear. For example, in lines 52-53/148-149 in the Introduction Section, the authors claim that previous methods used "single transitions or sequences of transitions from the same episode" to encode the context and state that incorporating cross-episodic information is one of the novelties of the proposed method. This claim is false as many previous works (e.g. VariBAD, TrMRL...) encoded transitions from multiple episodes into a single representation.

3) The empirical study, which is shown in Section 5, is not sufficient to support the claims of the authors (mainly hypothesis 1 and 3):  \
    a) More benchmarks including ones from the original papers (e.g. VariBAD, RL2) are missing. It is not clear if the results will hold on benchmarks where sufficient hyperparameter tuning of the other baselines took place

    b) An ablation study is missing, especially over the disentangled transformer architecture

    c) Regarding hypothesis 3 - the authors did not compare to theoretical/empirical line of works in generalization in Meta-RL (e.g [1,2])

[1] Lee et al. Improving generalization in meta-rl with imaginary tasks from latent dynamics mixture \
[2] Rimon et al. - Meta Reinforcement Learning with Finite Training Tasks -- a Density Estimation Approach

**Questions:**

1. In the title of the paper you suggested the method is "Efficient", but did not mention in what sense (not in the introduction at least). Do you mean in terms of time complexity (as discussed in lines 267-274)? If so, mention it in the introduction and add some results showing latency/FLOPs compared to "standard" (line 268) approaches.
2. In line 211 you mentioned you sample a sequence of $T$ transitions from each episode.\
    a) How do you sample the transitions? Uniform sampling? With replacement?\
    b) What is the chosen value for the different benchmarks? Did you test the effect of this value on the runtime/performance?\
    c) DId you also sample the transitions when running the other baselines?
3. Why did you choose to add positional encoding to the IET?
4. What are the number of episodes, length of each episode, and $T$ in each benchmark?
5. Did you use the same hyperparameters for all benchmarks? If not, how sensitive is your method to different hyperparameters?
6. How did you choose the hyperparameters for the baselines?
7. How many seeds did you use for the experiments in Section 5?
8. In Figure 5 the results of TrMRL don't match the ones in the TrMRL paper. Do you have an explanation for this? Did you run this baseline as done in the original paper (connecting to questions 2c and 6)?
9. How many learnable parameters does your architecture have compared to the other baselines (especially TrMRL)?
10. Typo line 273: ihas -> has

---

> ### Author Response · Authors · 2024-11-23
>
> We appreciate the reviewer's inquiries and have taken steps to address them thoroughly in our updated manuscript. We have uploaded our updated manuscript which incorporates your feedback.
>
> ---
>
> **TrMRL Results**
> The TrMRL paper focuses on **meta-training tasks** (e.g., "MetaWorld-ML45-Train"), where the tasks are already seen during training. Citing from **Section 5.2** in their paper:
>
> > "Finally, for ‘MetaWorld-ML45-Train’ (a simplified version of the ML45 benchmark where we evaluate on the training tasks), the more complex environment in this work, TrMRL achieved the best learning performance among the methods, supporting that the proposed method can generalize better across very different tasks in comparison with other methods."
>
> This setup inherently results in higher performance.
> Our evaluation emphasizes **meta-testing tasks**, which assess generalization to unseen environments. This aligns with the core focus of our work—generalization beyond the training tasks.
>
> When we compare our TrMRL results specifically in the **meta-training context**, they are consistent with those reported in the TrMRL paper.
>
> ---
>
> **Regarding the positioning of our approach**
>
> Thank you for pointing out the misrepresentation in our original phrasing regarding prior work that encodes transitions from multiple episodes. We appreciate the opportunity to clarify the novelty and positioning of our approach. Based on your feedback, we have revised the introduction to provide a more accurate positioning of our work relative to existing methods such as VariBAD, TrMRL, and AMAGO. Below, we provide a concise summary of the unique positioning of our approach:
>
> 1. *Acknowledgment of Prior Work*: We recognize and detail the contributions of prior works like VariBAD, TrMRL, and AMAGO in encoding transitions across episodes. This provides an accurate context for understanding the advancements our method offers.
> 2. *Specificity of Our Contribution*:
> - Unlike VariBAD and RNN-based methods, our transformer architecture effectively handles long-term task dynamics across multiple episodes without being hindered by vanishing gradients or memory constraints.
> - Compared to TrMRL and AMAGO, our method achieves a better balance of computational efficiency and performance by reducing runtime complexity while maintaining the ability to process cross-episode information. Further, our hierarchical design using the IET and CET hierarchically enables us to bias the learning to explicitly make use of intra-episodic information. In flat architectures (such as TrMRL) this has to be learned. Depending on the available data, this might be far from trivial to learn, even if it might be beneficial.
> 3. *Practical Relevance*: Our work addresses the computational and generalization challenges that have limited the applicability of Meta-RL methods in real-world settings, such as robotics and dynamic environments. Through empirical evaluations, we show that our method achieves faster convergence during meta-training while requiring fewer timesteps to generalize effectively to out-of-distribution tasks. This is especially evident in our comparisons with baselines such as TrMRL and AMAGO, where our approach consistently outperforms them in both **training efficiency** and **final generalization** performance. These results underscore the practical utility of our method in applications where computational resources and time are critical constraints, making it a more viable choice for real-world deployment.
>
> We believe these changes clarify the novelty and positioning of our work while highlighting its advantages over existing methods. The updated introduction now accurately conveys our contributions and how they advance the state-of-the-art in Meta-RL.

---

> ### Author Response · Authors · 2024-11-23
>
> **Additional Benchmarks and Baselines**
>
> To address the concern regarding benchmarks and baseline comparisons, we have added the following:
>
> 1. *New Benchmarks*:
>
>    - We included locomotion tasks such as **HalfCheetahVel**, **HalfCheetahDir**, and **AntDir** to evaluate the generalization and meta-training capabilities of our approach.
>    - **Figure 4** shows the comparison in terms of the generalization capabilities at test time, while **Figure 12** presents the meta-training curves for these new benchmarks. Our method, together with TrMRL and VariBAD shows a robust performance across the three tasks.
>
>
> 2. *New Baseline*:
>
> - We included AMAGO as an additional baseline for the ML10 and ML45 benchmarks in Figure 6.
> - This allows for direct comparisons with a strong transformer-based Meta-RL method capable of processing long sequences of transitions across episodes.
> - These additions enable a broader evaluation of our approach across diverse benchmarks and a rigorous comparison against a competitive state-of-the-art baseline.
> - Results show that ECET achieves superior performance compared to AMAGO, highlighting the effectiveness of our disentangled transformer architecture in capturing task representations efficiently.
>
>
> ---
>
> **Ablation Study**
>
> We recognize the importance of an ablation study to isolate the contributions of our transformer architecture (ECET) and its components. In fact, we already provide an extensive ablation study in **Section A.6.5 of the appendix**, where we systematically analyze the contributions of individual components. We have also added a more comprehensive discussion of the ablations in the main paper in Section 5.3. For the reader's convenience, this study is summarized as follows:
>
> 1. *Impact of Key Hyperparameters*:
>
> - We varied the number of episodes ($E$) in memory and the sequence length ($T$) of sampled transitions, as shown in Figure 28.
> - Results show that our chosen configuration of $E=25$ and $T=5$ balances fast convergence and high performance, while extreme values (e.g., $E=50$, $T=50$) result in suboptimal or slower improvement due to inefficiencies in representation learning.
>
> 2. *Sampling Strategy*:
>
> - We compared sampling one transition sequence per episode (our default) against sampling multiple sequences with replacement across episodes in Figure 29 (top plot).
> - Results indicate that sampling from episodes individually without replacement leads to faster and more reliable improvements, particularly for smaller $E$ and $T$ values.
>
> 3. *Policy Input Augmentation*:
>
> - We analyzed the effect of augmenting the policy $\pi$ with a linear transformation of the state versus relying solely on the transformer’s output in Figure 29 (middle plot).
> - Results show a significant improvement when including the linear transformation, supporting its inclusion in our method.
>
> 4. *Architectural Variations*:
>
> - We ablated the positional encoding in the Cross-Episode Transformer (CET) and simplified our model to a flat architecture similar to TrMRL, as shown in Figure 26.
> - Positional encoding in CET leads to suboptimal convergence due to misleading ordering in representations. The flat architecture struggles to capture comprehensive task representations, highlighting the effectiveness of our hierarchical approach.
>
> 5.*Comparison with Extended Context TrMRL*:
>
> - We extended TrMRL to process sequences of the same length as ECET ($E \times T$) in Figure 27.
> - Results show that TrMRL with extended context faces significant inefficiencies in extracting task meta-features, highlighting the advantages of ECET's hierarchical design for encoding shorter sequences across episodes.
>
> ---
>
> **Clarification of Hypotheses and Claims**
> 1. *Hypothesis 1 (Generalization Capabilities)*:
>
> - The robust test performance in the newly added MuJoCo benchmark in Figure 4 further strengthens our claim that ECET improves generalization.
> - Ablation studies further demonstrate that our context sampling strategy and hierarchical architecture are critical to this improvement.
>
> 2. *Hypothesis 3 (Efficiency and Robustness)*:
>
> - Comparisons with TrMRL and TrMRL (extended context) in Figure 27 show that ECET achieves superior meta-training efficiency and anytime performance, even when processing equivalent context lengths.
> - Ablations of architectural components confirm that ECET’s disentangled transformer design balances computational efficiency with robust task representation learning.
>
> ---
>
> We hope these changes and clarifications address your concerns and demonstrate the robustness of our study. Thank you again for your insightful comments, which have significantly strengthened our work.

---

> > ### Author Response · Authors · 2024-11-23
> >
> > **Experimental Setup**
> > We used the following experimental setup. We also went over our experiments section again and made sure that this information is explicitly stated.
> > - In our experiments, we trained the methods for \(10^{7}\) steps in the MuJoCo environments, \(5 \times 10^7\) steps in Meta-World, and \(2.5 \times 10^7\) steps in ManiSkill.
> > - Each experiment was repeated with 5 random seeds to ensure robustness, and we report the mean and standard error of the performance metrics in the plots.
> > - The episode lengths for each benchmark were as follows:
> >   - **MuJoCo:** Maximum of 200 steps.
> >   - **MetaWorld:** Maximum of 500 steps.
> >   - **ManiSkill:** Maximum of 50 steps.
> >
> > ---
> >
> > **Learnable Parameters**
> > - The total number of learnable parameters in our architecture (ECET) is **58,736**, compared to **58,280** in TrMRL. Despite the minimal difference in parameter count, ECET achieves significant improvements in generalization and training efficiency, validating the architectural enhancements.
> > - We added the hyperparameters that we use for our method in **Table 1 in Appendix A.3**.
> > - For TrMRL, we use the hyperparameter values suggested by the authors in their code repository. We use the same hyperparameter configuration in MetaWorld and ManiSkill, the same as we do with our method.
> >
> > ---
> >
> > We appreciate your detailed comments, which have allowed us to refine our methodology and provide more transparency in our results. We hope this response addresses your concerns and provides a clearer understanding of our experimental setup and findings.

---

> > > ### Comment · Reviewer_URVA · 2024-11-24
> > >
> > > I appreciate the authors' clarification, added experiments and discussions, and the overall effort they made in the rebuttal.
> > >
> > > I do believe some of my questions remain unanswered, e.g. Q1,Q2,Q3. I hope I am not mistaken, it was quite difficult to keep track of the authors' responses as they did not number the answers based on the question numbers and did not mark the changes in the new revision.
> > >
> > > Regarding the novelty and positioning of the work (weaknesses #1,2): I appreciate the clarification and the modifications made to the introduction and prior work. I think these highly improve the positioning of the proposed method.
> > > 1. Comparison to Varibad and RNN based methods - there are ways to mitigate the “vanishing gradients or memory constraints” of RNNs, e.g. TBPTT (truncated back propagation through time), which I believe was done in Varibad for example. Why should your method be superior in terms of memory consumption? Do you have any results to support this claim?
> > > 2. “enables us to bias the learning to explicitly make use of intra-episodic information” - can you explain what you mean by that?
> > > 3. “These results underscore the practical utility of our method in applications where computational resources and time are critical constraints, making it a more viable choice for real-world deployment” - did you perform latency/FLOP comparison between your method and other baselines to support this claim? There is an inherent advantage to RNN-based methods (e.g Varibad) in terms of runtime, so the results here are non-trivial in my opinion.
> > > 4. Regarding generalization - I still think a theoretical or at least a more rigorous discussion, explaining why this architecture should help generalization while comparing to previous works in generalization in Meta-RL is missing. Without this, I still find the novelty of the paper to be limited, as hierarchical transformers have been already extensively studied by prior works.

---

> ### Author Response · Authors · 2024-11-25
>
> ---
>
> **Thank you for your thoughtful feedback and for engaging in discussions to help us improve our work.**
> We sincerely apologize for any difficulty in tracking our responses due to the lack of explicit numbering. Below, we address your latest comments and the remaining concerns systematically.
>
> ---
>
> ### **Q1: Comparison to VariBAD and RNN-based methods**
> - **TBPTT and Its Trade-offs:**
>   While TBPTT (Truncated Backpropagation Through Time) can mitigate vanishing gradients and memory constraints in RNNs, it introduces certain trade-offs:
>   - TBPTT requires manually defined truncation intervals, which may hinder long-term dependency learning, especially in tasks with sparse rewards or delayed returns.
>   - This manual truncation can cause a loss of information critical for tasks requiring far-reaching temporal dependencies.
> - **Advantages of Transformer Architectures:**
>   - Vaswani et al.(2017) showed that transformer-based architectures inherently avoid these limitations by processing sequences in parallel, while also being well-suited for capturing long-term task dynamics.
> - **Empirical Evidence:**
>   - The key difference between ECET and VariBAD lies in the task encoding mechanism: ECET uses a two-tiered transformer architecture, whereas VariBAD relies on an RNN. Both methods share RL2 as the base meta-RL algorithm.
>   - Our experimental results consistently demonstrate that ECET outperforms VariBAD, underscoring the benefits of our approach on efficiency in terms of meta-training timesteps and long-term dependency handling.
>
> ---
>
> ### **Q2: Explicit Use of Intra-Episodic Information**
> - **Architectural Rationale:**
>   - Our disentangled architecture explicitly separates intra-episodic and cross-episodic representations using two distinct transformer components:
>     - **IET (Intra-Episodic Transformer):** Captures task-specific behaviors within an episode.
>     - **CET (Cross-Episodic Transformer):** Aggregates meta-level task characteristics across episodes.
> - **Empirical Justification:**
>   - As shown in our ablation studies (Figure 26 in Appendix A.6.5), flattening the architecture (as in TrMRL) to process both types of dependencies in a single transformer leads to poorer generalization. This result validates the architectural choice to disentangle these processes.
>
> ---
>
> ### **Q3: Latency/FLOP Comparison and Practical Utility**
> - **Focus on Meta-Training Efficiency in Terms of Timesteps:**
>   - Our claims of practical utility focus on the **efficiency gains in meta-training timesteps** achieved by the disentangled transformer architecture. This stands in contrast to approaches that rely on a single transformer layer to process long sequences spanning multiple episodes. Importantly, we did not intend to overclaim practical applicability. Our primary claims relate to the performance comparison of ECET with TrMRL and AMAGO, as demonstrated in the meta-RL setting.
> - **Clarification of Claims:**
>   - We acknowledge that the statement regarding the practical utility of ECET, as presented in the rebuttal, may have been interpreted as an overclaim. To address this, we have already revised the corresponding statement in the rebuttal response to all reviewers to ensure clarity.
>   - Importantly, this claim does not appear in our manuscript, as we have not made any assertions about runtime or FLOP superiority compared to RNN methods there. To further reinforce this point, we have now explicitly stated in the introduction of the manuscript that our primary claims relate specifically to **meta-training timestep efficiency**, as demonstrated in our results. That said, we do have superiority in terms of runtime complexity compared to the other transformer-based baselines, and we clarify that in the end of Section 4 in our manuscript.
> ---

---

> ### Author Response · Authors · 2024-11-25
>
> ### **Q4: Generalization and Novelty of the Architecture**
> - **Conceptual and Empirical Basis for Generalization:**
>   - Vaswani et al. (2017) highlights the transformer’s ability to model long-range dependencies through self-attention mechanisms, making it effective for sequence modeling tasks. This conceptual advantage, combined with our disentangled  intra- and cross-episodic dependencies, supports our method's capacity to generalize.
>   -  While approaches using multiple transformers in a hierarchy have been proposed (Shang
> & Ryoo, 2021; Correia & Alexandre, 2022; Mao et al., 2023; Zhang et al., 2023), we are the first to propose a hierarchical architecture that through capturing intra- and cross-episodic meta-features aims to process long sequences more efficiently in terms of runtime complexity:
>
>         - Shang & Ryoo (2021) use hierarchical architectures for offline RL, where the first tier (Step Transformer) transformer encodes a single observation (s, a, r) step, whereas the second tier (Sequence Transformer) encodes the sequence of Step Transformer embeddings.
>         - Correia & Alexandre (2022) use hierarchical transformers for offline RL, where the first tier (High Level Mechanism) processes states to generate subgoal embeddings, whereas the second tier (Low Level Mechanism) processes states, actions and subgoals to generate the next action.
>         - Mao et al. (2023) use a Vision Transformer in the first tier to encode the observations instead of a CNN, whereas the second tier encodes the sequence of the Vision Transformer embeddings.
>         - Zhang et al. (2023) use hierarchical transformers for multi-agent systems. They use the first tier (Inner Transformer Block) to encode individual agent states, whereas the second tier (Outer Transformer Block) encodes the sequence of all agent states and previous actions.
>
>     - These works, as well as TrMRL and AMAGO as transformer-based baselines from our paper, when processing sequences of transitions of length $T$ spanning multiple episodes $E$, have a runtime complexity O\($E^{2}\times T^{2}$\) whereas ECET, due to its architecture, has a runtime complexity O\($E^{2}+T^{2}$\).
>
>     - Empirical evidence from our ablation studies (e.g., Figure 26) further demonstrates that our architectural choices improve generalization compared to flat architectures.
>
> - **Relation to Task Augmentation Methods:**
>     - The referenced works (Lee et al., 2021; Rimon et al., 2022) address task augmentation in meta-RL, a different approach from task inference, which is the focus of our method.
>         - **Lee et al. (2022):** Uses task augmentation with imaginary tasks from a latent dynamics mixture to enhance generalization.
>         - **Rimon et al. (2022):** Samples tasks from a KDE to augment the training set and improve the diversity of the task distribution.
>   - These methods are complementary to our approach and could be integrated with ECET to further enhance its performance. We have added a discussion of these works in the Related Work section and highlighted task augmentation as a promising direction for future work.
>
> ---

---

> > ### Author Response · Authors · 2024-11-25
> >
> > ---
> >
> > ### **Responses to Q1, Q2, and Q3 from the Initial Review:**
> >
> > **Q1: Efficiency of ECET**
> > - By describing our method as "efficient," we specifically refer to **efficiency in terms of meta-training timesteps**. Unlike methods that require extensive training steps to converge, ECET achieves better performance with fewer meta-training timesteps, as demonstrated in our results.
> > - We acknowledge that this was not clearly articulated in the initial manuscript, and we have since amended the introduction to explicitly clarify this point. This revision ensures that readers understand the scope and context of the efficiency claim, which focuses on meta-training timesteps rather than other aspects such as runtime or FLOPs.
> >
> > ---
> >
> > **Q2: Sequence Sampling and Comparisons with Baselines**
> >
> > - **Sequence Length (T):**
> >   - We varied the number of episodes ($E$) in memory and the sequence length ($T$) of sampled transitions, as shown in Figure 28 in Appendix A.6.5.
> > - Results show that our chosen configuration of $E=25$ and $T=5$ balances fast convergence and high performance, while extreme values (e.g., $E=50$, $T=50$) result in suboptimal or slower improvement due to inefficiencies in representation learning.
> >
> > - **Sequence Sampling Approach:**
> >   - We compared sampling one transition sequence per episode (our default) against sampling multiple sequences with replacement across episodes in Figure 29 (top plot) in Appendix A.6.5.
> > - Results indicate that sampling from episodes individually without replacement leads to faster and more reliable improvements, particularly for smaller $E$ and $T$ values.
> >
> > - **Comparison with Baselines:**
> >   - Other baselines follow fundamentally different approaches to handling sequences:
> >     1. **MAML PPO:** Uses only the current state as input, without considering past transitions.
> >     2. **PEARL:** Samples multiple individual transitions from a buffer of past experiences.
> >     3. **RL2, VariBAD, and AMAGO:** Process multiple episodes in sequence.
> >     4. **TrMRL:** Uses a sequence of the five most recent transitions as input.
> >   - We evaluated these baselines without altering their core algorithms to ensure a fair comparison.
> >
> > ---
> >
> > **Q3: Importance of Positional Encoding**
> > - Transformers, by design, excel at processing sequences without requiring sequential training. However, to handle sequences effectively, they rely on **positional encodings** to preserve information about the ordering of the elements in the sequence (Vaswani et al., 2017).
> > - For our model, positional encodings are used in the **IET (Intra-Episodic Transformer)** component to encode transition sequences within an episode. Without positional encoding, the information on the order of transitions would be lost, which is critical for learning meaningful intra-episodic dependencies.
> > - Conversely, positional encodings are **not used in the CET (Cross-Episodic Transformer)**, as the ordering of episodes is not meaningful in on-policy algorithms. Episodes in this setting are collected with the same policy, making their sequence arbitrary.
> > - We ablated the use of positional encoding in CET (Figure 26 in Appendix A.6.5) and observed no significant impact on performance, further validating this design choice.
> >
> > ---
> >
> > **We deeply appreciate your constructive feedback and the opportunity to address your concerns.** Your comments have been instrumental in improving the clarity and positioning of our work, and we are grateful for your engagement. Please feel free to let us know if further clarification or additional experiments are needed.
> >
> > ---
> >
> >  André Correia and Luís A. Alexandre. Hierarchical decision transformer. CoRR, abs/2209.10447, 2022.
> >
> > Suyoung Lee and Sae-Young Chung. Improving generalization in meta-RL with imaginary tasks from latent dynamics mixture. In Advances in Neural Information Processing Systems, 2021.
> >
> >  Hangyu Mao, Rui Zhao, Hao Chen, Jianye Hao, Yiqun Chen, Dong Li, Junge Zhang, and Zhen Xiao. Transformer in transformer as backbone for deep reinforcement learning. CoRR, abs/2212.14538, 2023.
> >
> > Zohar Rimon, Aviv Tamar, and Gilad Adler. Meta reinforcement learning with finite training tasks - a density estimation approach. In Advances in Neural Information Processing Systems, 2022.
> >
> > Jinghuan Shang and Michael S. Ryoo. Starformer: Transformer with state-action-reward representations.CoRR, abs/2110.06206, 2021.
> >
> > Ashish Vaswani, Noam Shazeer, Niki Parmar, Jakob Uszkoreit, Llion Jones, Aidan N. Gomez, Ł ukasz Kaiser, and Illia Polosukhin. Attention is all you need. In Advances in Neural Information Processing Systems, 2017.
> >
> >  Bin Zhang, Hangyu Mao, Lijuan Li, Zhiwei Xu, Dapeng Li, Rui Zhao, and Guoliang Fan. Stackelberg decision transformer for asynchronous action coordination in multi-agent systems. CoRR, abs/2305.07856, 2023.

---

> > > ### Author Response · Authors · 2024-11-29
> > >
> > > Thank you for taking the time to review our paper. If you have any additional questions on the rebuttal or require further clarification, we would be glad to assist. Please feel free to reach out at any time.

---

### Official Review · Reviewer_khht · 2024-10-29

**Soundness:** 3
**Presentation:** 3
**Contribution:** 2
**Rating:** 6
**Confidence:** 5

**Summary:**

This paper proposes a new algorithm; Efficient Cross-Episodic Transformers (ECET) for online meta-reinforcement learning. Their method tackles the problem of enabling reinforcement learning agents to perform effectively in previously unseen tasks. In order to achieve that, they take past episodes and use them as in-context information, considering different task variations and single transitions by not only extracting information from within sequences of transitions but also incorporating information across sequences sampled from different episodes; thus, integrating insights from prior meta-RL research. Their main contribution is improving the generalization capabilities and providing more efficient meta training, outperforming previous state-of-the-art methods. They perform an empirical study in the Meta-World and Maniskill Benchmarks to conclude that their extended algorithm captures more general task features and outperforms the state-of-the-art in online meta-RL algorithms in few-shot adaptation to parametric variations of tasks and out-of-distribution tasks.
Overall, this paper could be a significant algorithmic contribution, with the caveat of some clarifications on the theory and experiments. Given these clarifications in an author's response, I would be willing to increase the score.

**Strengths:**

For originality, the paper presents a novel method for learning meta-features through efficient cross-episode meta-RL, using a novel transformer architecture that enhances the adaptability of RL agents across diverse sets of tasks. For significance, their approach leverages intra-episode experiences and cross-episode experiences, providing a more comprehensive learning process and integrating ideas from previous meta-RL research. From a perspective of clarity and quality, their learned algorithm is capable of outperforming the previous state-of-the-art and providing more efficient meta-training while significantly improving generalization capabilities. In general, the proposed method enhances the agent’s ability to generalize from limited data supported with sufficient empirical experiments and paves the way for more robust and versatile AI systems.

The paper has some good points to highlight and summarize:
- Improving generalization capabilities and providing efficient meta-training.
- Outperforming previous state-of-the-art methods by capturing complex, multilevel patterns in the data.
- Integrating multiple ideas from previous meta-RL research.

**Weaknesses:**

Based on my experience, while I truly appreciate the authors’ effort, I believe that their paper lacks novelty and can be considered as an improvement but not as a real contribution for several reasons. First of all, there is no central contribution beyond improving generalizing capabilities by introducing a new architecture and outperforming previous SOTA–which was supported in the empirical results in figures 4, 5, and 6— still, their method’s generalization to tasks that are completely different from the meta-training set remains limited as their approach is tailored to exploiting similarities as shown in hypothesis 3: ECET outperforms the state-of-the-art online meta-RL in out-of-distribution (OOD) tasks. This leads to Section 1, Related Work, it is still unclear how their method differs from other baselines in solving the same problem, specifically the TrMRL method Melo (2022), which is the closest to the authors’ proposed algorithm. In other words, the authors’ addition and value are still vague compared to previous research tackling the same research question. Having read Melo (2022), I see that their cross-episode experience equals the episodic memory system proposed in the TrMRL method. More discussion on these topics would be helpful.


Minor comments:
1. Page 6: Line 4, ‘ihas’ should be corrected.

**Questions:**

For the theory, there are a few steps that need clarification and further explanations on novelty. For novelty, it needs to be clarified if their method is being stated as a novel result. It looks like the method has already been shown in "Transformers are Meta-Reinforcement Learners.” There is a statement that “TrMRL is the closest baseline to our proposed method (Melo, 2022). It uses a transformer in the inner loop of the RL2 algorithm. It takes a sequence of the 5 most recent transitions as input to the transformer. We use the implementation provided by the authors.” Is your algorithm 2 somehow an extension? Is your theorem 3 completely new?

In Appendix A.2.3 Procedural Components, you mentioned that “SampleTransitions: We sample the transition sequences of length T from each episode in H. For past episodes, we randomly” You randomly did what?

There is also one concept in the algorithm that I cannot verify. How did you apply the intra-episode and cross-episode simultaneously? More about this would be illuminating

For the experiments, the following should be addressed:
- The central contribution is enhancing the generalizing capabilities. The authors concluded that their method is capable of adjusting to completely unseen tasks that share the same state and action spaces as the training tasks. Despite outperforming state-of-the-art methods in online meta-RL by efficiently compressing cross-episodic knowledge, their method’s generalization to tasks that are completely different from the meta-training set remains limited as it is tailored to exploiting similarities. I would like to see some discussions on this and I think it would be beneficial to demonstrate that empirically.

---

> ### Author Response · Authors · 2024-11-23
>
> Thank you for your feedback and for raising these important concerns. We value the opportunity to clarify the novelty and contributions of our work, particularly in relation to TrMRL (Melo, 2022) and its memory system, as well as the broader impact of our approach.
>
> ---
>
> **Novelty and Central Contribution**
>
> While our approach indeed builds on prior work, we believe it introduces the following **central contributions** that go beyond incremental improvements:
> 1. **Disentangled Transformer Architecture:**
>    - Our method employs a hierarchical transformer-based architecture (ECET) that disentangles intra- and inter-episode representations. Specifically:
>      - **Intra-Episode Transformer (IET):** Encodes short sequences of transitions within individual episodes.
>      - **Cross-Episode Transformer (CET):** Aggregates these intra-episode representations across multiple episodes to capture global task characteristics.
>    - This disentanglement enables ECET to handle both short-term dynamics and long-term cross-episode task patterns more effectively than TrMRL, which uses a monolithic transformer that processes sequences without distinguishing between intra- and inter-episode information.
>
> 2. **Efficiency in Handling Context:**
>    - TrMRL's episodic memory system uses a single transformer to process sequences, limiting its capacity to scale to longer sequences due to runtime complexity. Our hierarchical design allows for shorter sequences to be processed in IET before aggregation in CET, reducing computational overhead and improving training efficiency in terms of timesteps.
>    - This advantage is particularly evident in **out-of-distribution (OOD) tasks**, as our method generalizes more effectively to unseen task variations while maintaining computational efficiency (Figures 4, 5, 6, 7).
>
> 3. **A Broader Scope for Cross-Episode Experiences:**
>    - Unlike TrMRL, ECET explicitly integrates information from multiple episodes to build a more comprehensive task representation. This approach is critical for tasks requiring adaptation to nuanced or complex inter-episode relationships. In other words, TrMRL is well poised to identify the task dynamics, which can help to adapt the policy to improved performance. However, by only looking at the dynamics within a single episode TrMRL is unlikely to capture how the previous policies rollout relates to the updated policies rollout. Thus TrMRL will not be able to exploit previous policy behavior in areas where it was good or avoid it explicitly where it was bad. Our approach on the other hand can identify the task as well as how the policy/policies differ across rollouts. Overall we would argue that our approach thus gets us much closer to in context learning to learn by reinforcement as we capture environment dynamics, coupled with learning dynamics.
>
> ---
>
> **Generalization Beyond Similar Tasks**
>
> We agree that generalization to tasks completely different from the meta-training set remains challenging. However, we emphasize that our method outperforms state-of-the-art baselines, including TrMRL and AMAGO (see also response to Reviewer ixfQ), on out-of-distribution tasks. While our approach leverages similarities across tasks (as stated in Hypothesis 3), this is an inherent limitation of most meta-learning methods, including TrMRL. Importantly:
> - Our method demonstrates improved **performance across timesteps**, faster meta-training convergence and improved meta-test performance, enabling agents to generalize to tasks with greater variations.
> - The inclusion of **diverse benchmarks** (Figures 4, 5, 6, 7) showcases our method’s robustness across task distributions with varying degrees of similarity to the meta-training set.
>
> We acknowledge this limitation and view it as a promising direction for future work to extend ECET’s capabilities to handle tasks with minimal or no similarities to the training distribution.

---

> > ### Author Response · Authors · 2024-11-23
> >
> > **Relation to TrMRL (Melo, 2022)**
> >
> > Key distinctions to TrMRL include:
> > 1. **Architectural Design:**
> >    - TrMRL employs a single transformer to process sequences of transitions, treating all information as part of a single sequence.
> >    - In contrast, ECET disentangles intra-episode (short-term) and inter-episode (long-term) information, explicitly addressing the hierarchical nature of task representations.
> >
> > 2. **Context Sampling Strategy:**
> >    - TrMRL uses flat sequences sampled from transitions, while ECET introduces a hierarchical context sampling mechanism, enabling efficient aggregation of information across episodes as depicted in Figure 2.
> >
> > 3. **Scalability and Generalization:**
> >    - ECET handles longer sequences and larger episode sets more efficiently, as shown in our ablation studies (Figures 26-29). This enables better performance on OOD tasks, whereas TrMRL’s performance degrades in terms of performance across timesteps when attempting to process longer sequences due to its flat architecture.
> >
> > These differences highlight the architectural and methodological advances of ECET, which are critical for tackling the challenges associated with generalization in Meta-RL.
> >
> > ---
> >
> > We appreciate the feedback that our contribution compared to TrMRL may have been unclear. To address this, we revised the **Introduction** and **Related Work** sections to explicitly discuss how our approach differs from TrMRL and other baselines tackling similar research questions.
> >
> > We thank the reviewer for their thoughtful comments, which have helped us refine the positioning of our work and clarify its contributions. We hope these revisions address your concerns and better demonstrate the novelty and value of our approach.

---

> > ### Comment · Reviewer_khht · 2024-11-26
> >
> > I would like first to thank you for your detailed clarifications regarding differences from the TrMRL. I am going through your revisions and will update my score accordingly.

---

> > > ### Author Response · Authors · 2024-11-30
> > >
> > > Thank you for your thoughtful feedback and for reviewing our rebuttal. We would be happy to answer any further questions or clarify any remaining concerns you might have. Please don't hesitate to reach out.

---

### Official Review · Reviewer_iQfe · 2024-11-02

**Soundness:** 4
**Presentation:** 4
**Contribution:** 3
**Rating:** 8
**Confidence:** 4

**Summary:**

This paper proposes a method for learning features for meta-reinforcement learning in latent space by training a two-level hierarchy of transformer networks. The lower layer is trained to represent trajectories experienced while doing a specific task (i.e. experiencing a single MDP) while the upper layer is trained on all tasks. This method is shown to have significantly better performance than several other meta-RL methods on a variety of meta-RL benchmarks, with the primary hypothesis for this being that the proposed architecture allows the networks to learn more general features that are informative across a variety of tasks. This hypothesis is supported by some experiments showing that embedded representations learned by the proposed method tend to cluster together by task more than representations learned by other methods.

**Strengths:**

Paper is overall clearly written and explained.

Related work positions the paper well.

Experiments are well designed.

I am a big fan of explicit hypothesis statements, and very much appreciate this paper for being clear about what it is investigating.

The idea is relatively simple and seems like a clear opportunity for more research.

The performance is impressive and the results are quite thorough.

**Weaknesses:**

There are few weaknesses in my opinion. Instead, I will give some minor feedback.

- “By considering both intra-episode experiences (information within a single episode) and cross-episode experiences (information across multiple episodes), our method can learn richer meta-features for the environment.” The opposite of intra is inter, not cross. The title already contains 'cross', but consider using this term when explaining (I believe it already occurs once towards then end).

- Consider citing Bhatia et al. (RL^3: Boosting Meta Reinforcement Learning via RL inside RL^2) which also uses transformers and learns state-augmentation features for meta-RL, albeit not through inter-episode data sharing.

- Figure 2 bleeds into the margins.

- It could be clearer which baselines (or other meta-RL methods) this method is compatible with and which are mutually exclusive.

**Questions:**

“Hypothesis 1: ECET captures more general task features through its ability to capture intra- as well as cross-episodic experiences.”

I recognize claims about feature generality are difficult to substantiate, and that it is somewhat of a catch-22 to try to deal with these topics experimentally. I appreciate the authors for making an attempt. I am interested in their thoughts regarding essentially more rigorous versions of such experiments.

- Do you have a definition for the generality of a task feature? How does one measure this?

- You’ve shown that embedding vectors tend to cluster for some tasks. Do you think there are any possible confounders beyond generality of features?

- Moreover, if one were to take a more systematic and focused approach to quantifying the difference between tasks so that embeddings generated from those tasks may be compared to the embedding distances --- similar to what is presented in figure 13 but with some notion of how similar two columns/rows ought to be --- do you suspect they would encounter anything surprising?

Is there anything that makes such an experiment difficult to execute other than initially deciding how to measure difference between tasks? I agree these results seem promising w.r.t. this hypothesis, but there are definitely a few more logical steps to nail down before this can be determined without doubt (which would be an important result in it own right, of course).

---

> ### Author Response · Authors · 2024-11-23
>
> Thank you for raising these insightful points and for recognizing the challenges inherent in experimentally substantiating claims about task feature generality. We appreciate the opportunity to elaborate on our thoughts regarding these aspects.
>
> ---
>
> **Definition and Measurement of Task Feature Generality**
>
> The concept of "generality" could benefit from a more nuanced consideration. While it's typically framed as the ability of task features to encode transferable task-relevant information, here is a deeper challenge: these features must also help inform how learning progresses, not just differentiate between tasks as we are following the RL2 paradigm. This introduces algorithm dependence into the notion of generality since the task features are tied to the algorithm used to generate the embeddings or data. As such, disentangling the contributions of task-specific features from those tied to algorithmic behavior is a significant challenge. Developing metrics that isolate and measure this pure notion of generality without conflating algorithmic influence might be particularly difficult and is worth discussing further.
>
> ---
>
> **Confounders in Embedding Clustering**
>
> We acknowledge that clustering embeddings for some tasks, as presented in **Figure 13**, may be influenced by factors beyond task feature generality. Potential confounders including
> limited data or noise in the sampled sequences that are used to generate task representations could lead to suboptimal clustering behavior.
>
> ---
>
> **Challenges in Systematic Experimentation**
> Concrete examples could clarify the challenges. For instance:
>
> 1. **Curating a Diverse Task Set**: Imagine generating task embeddings using multiple instantiations of our learning algorithm (i.e., with different hyperparameter configurations) across tasks with distinct dynamics (e.g., gridworlds, continuous control, and adversarial games). Ensuring diversity while maintaining meaningful comparisons across these two dimensions of variability would require careful and labor-intensive curation.
>
> 2. **Ground Truth for Task Differences**: Establishing ground truth becomes even harder when considering multiple configurations of the algorithm used to generate learning trajectories on different tasks. Each configuration might result in subtle differences in task difficulty or structure, which could lead to discrepancies in how tasks are represented. For example, a task embedding  generated using an instantiation with one hyperparameter setting that focuses on exploration might appear significantly different when the algorithm is reconfigured and less exploratory, even if the underlying task dynamics remain the same. Developing similarity metrics that remain robust across such algorithmic variability adds a layer of complexity to the already challenging task of defining task differences.
>
> 3. **Scalability**: Evaluating embeddings across tasks, and hyperparameter configurations could become prohibitively resource-intensive. This computational demand would grow exponentially if the diversity of tasks or algorithms increases, making systematic experimentation at scale a significant challenge.
>
>
>
> Despite these challenges, we believe that such systematic experimentation is an important direction for future research and could yield valuable insights into the nature of general task features.
>
> ---
>
> We appreciate your recognition of the promise shown by our results concerning Hypothesis 1. While our current work provides initial evidence for the hypothesis, we agree that additional logical steps, such as systematically linking task similarities to embedding distances, would significantly strengthen the claim. We see this as an exciting avenue for future research and thank you for your thoughtful suggestions, which will help guide this exploration.

---

> > ### Author Response · Authors · 2024-11-23
> >
> > **Terminology (Cross-Episode vs. Inter-Episode)**
> > To address the potential confusion regarding terminology, we have clarified the interchangeable use of "cross-episode" and "inter-episode" in the manuscript. Specifically, we added the following footnote:
> >
> > > "Throughout this paper, we use the term \emph{cross-episode} to describe interactions across multiple episodes, which is synonymous with \emph{inter-episode}. This choice aligns with the terminology used in the title for consistency."
> >
> > This ensures that the terminology remains consistent and understandable while aligning with the title of our paper. We also reviewed the manuscript to ensure uniform usage of these terms throughout.
> >
> > ---
> >
> > **Improving Related Work**
> > We have added a description of Bhatia et al. in the **Related Work** section. This paper is relevant as it proposes an extension of RL^2, which is the meta-RL algorithm we use for our method. However, the method goes in a different direction, injecting Q-values into R:^2. We highlight this distinction to better contextualize our contributions relative to RL^3 and ensure a comprehensive comparison.
> >
> > ---
> >
> > **Compatibility with Other Meta-RL Methods**
> > We have improved the Introduction and Related Work sections in the manuscript to clarify which meta-RL methods are compatible with our approach and which are not. Specifically:
> > - Our method is compatible with **transformer-based approaches** (e.g. TrMRL, AMAGO) and methods that can process sequential data (RL^2, VariBAD), as we share the architectural capacity to encode sequences effectively.
> > - It is not directly compatible with methods designed for fixed-length embeddings that do not rely on sequential processing (e.g., PEARL). .
> >
> > ---
> >
> > We appreciate your insightful comments, which have helped us refine the clarity, consistency, and positioning of our work. These changes ensure that our contributions are more clearly articulated and contextualized within the broader meta-RL literature.

---

> > > ### Comment · Reviewer_iQfe · 2024-11-25
> > > **Official Comment by Reviewer iQfe**
> > >
> > > Thank you very much for answering my questions and providing some clarification. I appreciate the effort put into the rebuttal and discussion phase. As my score is already positive, I am inclined to keep it so.

---

### Official Review · Reviewer_ixfQ · 2024-11-03

**Soundness:** 2
**Presentation:** 3
**Contribution:** 2
**Rating:** 6
**Confidence:** 4

**Summary:**

The paper proposes new online Meta-RL algorithm that is able to combine intra- and cross-episodic information to better make decisions under uncertainty. Each intra-episode is encoded into a latent variable by a transformer model, which are then processed altogether by another transformer and a context-specific latent is formed. Authors claim SOTA results on Maniskill and challenging Meta-World ML45 benchmarks, surpassing previous SOTA methods.

**Strengths:**

- The paper reports state of the art results on challenging tasks, including challenging environment Meta-World ML45.
- The method is explained clearly and visually explained.

**Weaknesses:**

[W1] Authors compare their method with TrMLR, which uses RNN in the outer-loop, while authors of the proposed method use transformer. I believe that the difference in the capacity can explain the performance distinction between two methods. To my mind, it is early to claim SOTA results without trying to match architectures first.

[W2] A strong baseline is missing from the paper: AMAGO [1]. The [1] authors do not report the results on ML45, stating it is left for the future work. However, since the main contribution of the paper is SOTA results, I believe AMAGO should be included in the baselines. Also, mind that AMAGO was introduced in ICLR”24.

[W3] Authors do not state whether the baselines (especially TrMLR) were tuned to get the best performance out of them. Again, as the paper claims SOTA results, it is vital to have this information.

**Questions:**

1. I would appreciate if authors include an additional baseline method on ML45, AMAGO [1], to ensure the result of the proposed method is indeed SOTA.
2. To justify SOTA performance, I would also like to see the experiments on TrMLR with transformer or mamba architecture as an outer-loop memory model.
3. I would also appreciate it if authors provide how many hyperparameters were swept for the baseline TrMLR method and which method for HPO was used.

Although my initial recommendation is to reject the paper, I would be happy to revise my score as the authors address my questions.

[1] Grigsby, Jake, Linxi Fan, and Yuke Zhu. "Amago: Scalable in-context reinforcement learning for adaptive agents.”

*After rebuttal update:* I changed my score to 6 since the questions were mostly addressed, see the comments for details.

---

> ### Author Response · Authors · 2024-11-23
>
> Thank you for your constructive feedback. We have incorporated your suggestions and made significant improvements to our study by adding new benchmarks, including AMAGO as a baseline, and clarifying the implementation details of TrMRL. Below, we provide a detailed account of the changes made to address your concerns.
>
> ---
>
> **On the Implementation of TrMRL**
>
> We appreciate your point regarding the architectural differences between TrMRL and our method. To ensure a fair comparison:
> - We used the **publicly available TrMRL implementation** and followed the exact pipeline proposed by its authors, as described in their code and paper.
> - TrMRL utilizes a **transformer in the inner loop** of the RL² algorithm. Our implementation makes no modifications to this setup.
> - **Hyperparameter Settings:**
>   -The authors of TrMRL provide the tuned hyperparameters for the MetaWorld and MuJoCo benchmarks, in the TrMRL repository.
>   - For Maniskill, we applied the same settings as in MetaWorld due to the absence of explicit guidance in their repository.
>   - For our method, we also used the exact same hyperparameter settings for MetaWorld and ManiSkill, detailed in **Table 1 (Appendix A.3)**. The only difference in hyperparameters comes from hyperparameters that are introduced for incorporating cross-episodic information.
>
> ---
>
> **On Comparing to AMAGO**
>
> Thank you for suggesting AMAGO as a baseline. We have now included **AMAGO** in our experiments and evaluated its performance on **ML10** and **ML45** benchmarks:
> - Results are presented in **Figure 6** and other figures summarizing ML10 and ML45 outcomes.
> - **Key Findings:**
>   - Transformer-based methods outperform non-transformer approaches.
>   - ECET achieves superior performance compared to AMAGO with at least twice as high a success rate in ML10 and ML45 each, highlighting the effectiveness of our disentangled transformer architecture in capturing task representations efficiently.
>
> By including AMAGO as a baseline, we broaden the scope of our evaluation and ensure that our results are positioned rigorously within the state-of-the-art.
>
> ---
>
> **Additional Benchmarks**
>
> To further address concerns about benchmarks and baseline comparisons, we have added the following:
>    - We included locomotion tasks such as **HalfCheetahVel**, **HalfCheetahDir**, and **AntDir** to evaluate the generalization and meta-training capabilities of our approach.
>    - **Figure 4** shows the comparison in terms of the generalization capabilities at test time, while **Figure 12** presents the meta-training curves for these new benchmarks. Our method, together with TrMRL and VariBAD shows a robust performance across the three tasks.
>
> ---
>
> **Conclusion**
>
> We are grateful for your thoughtful feedback, which has significantly strengthened our manuscript. By including new benchmarks, adding AMAGO as a baseline, and clarifying our experimental setup, we provide a more robust evaluation of ECET. These revisions ensure that our claims of SOTA performance are rigorously supported and transparently presented.
>
> Thank you again for your insightful comments, which have greatly contributed to improving the clarity and depth of our work.

---

> ### Comment · Reviewer_ixfQ · 2024-11-23
>
> I thank the authors for thoughtful rebuttal and additional experiments they provide. Before I revise my score, I have a question to ask: Figure 6 reports "Meta-Learning Timesteps" on x-axis. Is that the amount of iterations of gradient updates during training? If so, how many environment episodes were used to evaluate the models for the adaptation?

---

> ### Author Response · Authors · 2024-11-23
>
> Thank you for your question.
>
> In Figure 6, the "Meta-Learning Timesteps" on the x-axis represents the total number of individual timesteps accumulated across all meta-training tasks during the meta-learning process.
>
> Regarding the evaluation of the models during meta-test(adaptation), they were assessed using 25 episodes per task for MetaWorld and ManiSkill, and 5 episodes for the MuJoCo benchmark. This ensures a consistent and fair comparison of their performance in adapting to new tasks.
>
> We hope this clarifies your question and are happy to provide further details if needed.

---

> ### Comment · Reviewer_ixfQ · 2024-11-23
>
> I am leaning to change my score to 6 since authors addressed most of my questions, however I still have some concerns that, I believe, should be taken into account:
> 1.  To my mind, SOTA results should be backed up with a serious hyperparameter search of all the methods authors choose to compare to, not only the proposed one. However, I understand that it requires a large amount of resources. This is the main reason I do not revise my score above 6.
>
> 2. After the submission period of ICLR"25 had ended, there appeared a revised version of AMAGO-2 [1], which reported similar or superior results in Meta-RL setting and used more complicated environments to evaluate their method. At the same time, I know this paper should not be considered as a prior work during the review, but I still feel obliged to mention its existence here.
>
> 3. > Comparisons with TrMRL and AMAGO consistently show ECET’s superiority in both efficiency in terms of timesteps and final performance, highlighting its practical utility for resource-constrained real-world applications like robotics and dynamic environments.
>
>     Using transformers is a known challenge for the robotics. As mentioned in [2], a typical sampling rate of a robotic controller varies from 100Hz-1000Hz, which means 15K context length for an episode of 15 seconds. Following this, I would really refrain from such overclaims of practical utility of the proposed method without extensive evaluation first.
>
> [1] Grigsby, J., Sasek, J., Parajuli, S., Adebi, D., Zhang, A., & Zhu, Y. (2024). AMAGO-2: Breaking the Multi-Task Barrier in Meta-Reinforcement Learning with Transformers. arXiv preprint arXiv:2411.11188.
>
> [2] Schmied, T., Adler, T., Patil, V., Beck, M., Pöppel, K., Brandstetter, J., ... & Hochreiter, S. (2024). A Large Recurrent Action Model: xLSTM enables Fast Inference for Robotics Tasks. arXiv preprint arXiv:2410.22391.

---

> ### Author Response · Authors · 2024-11-23
>
> We appreciate your thoughtful feedback and thank you for your willingness to reconsider the score. Below, we address your concerns in detail:
>
> 1. **Hyperparameter Optimization**
>    We understand the importance of conducting a thorough hyperparameter optimization for all methods in comparative studies. However, as mentioned, our approach aligns with the hyperparameter settings used for TrMRL, as detailed in Table 1 in Section A.3 of the Appendix, with the exception of architecture-specific configurations for ECET. To clarify, **we did not perform any extensive hyperparameter optimization of our method**. While we acknowledge the importance of hyperparameter optimization (HPO), the resource requirements for meaningful searches across all methods are substantial. Nevertheless, we believe that this constraint does not undermine the key insights and contributions of our work, which focus on the efficiency and performance gains of ECET in a meta-RL setting.
>
> 2. **Comparison with AMAGO-2**
>    We sincerely thank you for bringing the revised AMAGO-2 paper to our attention. While we acknowledge its advancements in Multi-Task RL, it is important to note that its primary focus aligns more closely with Hypothesis 2 of our work. Specifically, AMAGO-2 primarily addresses generalization to parametric variations of tasks (e.g. more difficult levels) which exist in the meta-training set. This is also reflected in the fact that they don't test the performance of their approach in the meta-test set of the ML45 benchmark, instead they only show evaluation performance on ML45 meta-training tasks in Figure 4. This contrasts with our focus on broader generalization in meta-RL, including adaptation to entirely new tasks in the meta-test set.
>    Furthermore, the results reported for AMAGO-2 for ML45 (~0.6 Success Rate at 5e7 meta-training timesteps) are comparable to those we obtained with the original AMAGO. Based on our results with AMAGO, and Figure 4 from AMAGO-2, ECET consistently outperforms both in terms of performance on the meta-training set and efficiency at the same budget of timesteps. Thus, while AMAGO-2 makes valuable contributions to Multi-Task RL and generalization to parametric variations of tasks, its approach and problem formulation differ from those of our work. However, we do see it fitting to add it to our related works, and we have revised Section 2 of our manuscript accordingly.
>
> 3. **Practical Utility for Robotics**
>    We appreciate your comments regarding the applicability of transformers in robotics. We agree that deploying transformers in robotic systems poses significant challenges, as highlighted in [1]. However, our claims regarding the practical utility of ECET focus on the efficiency gains offered by the disentangled architecture we propose, which processes context windows more effectively. This stands in contrast to approaches that rely on a single transformer layer to process long sequences spanning multiple episodes.
> Importantly, we did not intend to overclaim practical applicability to robotics. Our primary claims relate to the performance comparison of ECET with TrMRL and AMAGO, as demonstrated in the meta-RL setting. To ensure clarity, we have amended the wording in our reply to all reviewers to prevent any potential misinterpretation of these claims. We appreciate your feedback on this and have taken steps to address it.
>
> ---
>
> We hope this addresses your concerns and provides clarity on the contributions and scope of our work. We remain grateful for your feedback and are open to further discussion.
>
> ---
> [1] Schmied, T., Adler, T., Patil, V., Beck, M., Pöppel, K., Brandstetter, J., ... & Hochreiter, S. (2024). A Large Recurrent Action Model: xLSTM enables Fast Inference for Robotics Tasks. arXiv preprint arXiv:2410.22391.

---

> ### Comment · Reviewer_ixfQ · 2024-11-23
>
> I understand the authors' position on the matter of hyperparameter optimization, and I do not question their academic integrity. As I understand it, one of the main contributions of the paper is claiming SOTA results; therefore, I believe it is vital to eliminate any ambiguity regarding the evaluation protocol. It may be that authors of, say, VariBAD or AMAGO did not exhaustively tune their hyperparameters, and one of the baseline methods might actually underperform because of it. I am quite pedantic about this matter, so I stand by my score.
>
> At the same time, I admire the authors' commitment to the rebuttal and their willingness to answer my questions. Thank you.

---

### Author Response · Authors · 2024-11-23
**Response to all reviewers**

Dear Reviewers,

We sincerely thank you for your constructive feedback and thoughtful comments. Your suggestions have been invaluable in improving the clarity, rigor, and scope of our manuscript. We have carefully considered each of your points and have made significant updates to address your concerns comprehensively. Below, we summarize the key changes and additions made to the revised manuscript, now available on OpenReview.

---

**On Including AMAGO as a Baseline**

We appreciate the suggestion to include **AMAGO** as a baseline. In response:
- We have evaluated **AMAGO** on both the **ML10** and **ML45** benchmarks. The results are presented in **Figure 6** and other figures summarizing ML10 and ML45 outcomes.
- **Key Findings:**
  - Transformer-based methods, including ECET and AMAGO, outperform non-transformer approaches.
  - ECET consistently achieves superior performance compared to AMAGO, underscoring the effectiveness of our disentangled transformer architecture in capturing task representations efficiently.

By adding AMAGO as a baseline, we strengthen the scope and rigor of our comparisons, ensuring our results are well-positioned within the state-of-the-art.

---

**On Additional Benchmarks**

To address concerns about the breadth of our evaluation:
- We added locomotion tasks such as **HalfCheetahVel**, **HalfCheetahDir**, and **AntDir** to further evaluate the generalization and meta-training capabilities of our approach.
- **Figure 4** compares generalization performance at test time across these tasks, while **Figure 12** illustrates the meta-training curves.
- Our results show that ECET, along with TrMRL and VariBAD, demonstrates robust performance across these tasks, further validating the versatility of our approach.

---

**On the Positioning of Our Approach**

We appreciate your feedback regarding the positioning of our method and its relation to prior work. In response, we have revised the **Introduction** and **Related Work** sections to clarify our contributions and distinguish our approach from existing methods such as VariBAD, TrMRL, and AMAGO. Key updates include:

1. **Acknowledgment of Prior Work:**
   - We detail the contributions of prior works like VariBAD, TrMRL, and AMAGO in encoding transitions across episodes. This provides a more accurate context for understanding the advancements offered by ECET.

2. **Specificity of Our Contribution:**
   - **Compared to VariBAD and RNN-based Methods:**
     Our transformer-based architecture effectively handles long-term task dynamics across multiple episodes, avoiding limitations like vanishing gradients and memory constraints that affect RNN-based approaches.
   - **Compared to TrMRL and AMAGO:**
     ECET achieves a better balance of computational efficiency and performance by reducing runtime complexity while maintaining the ability to process inter-episode information.

3. **Practical Relevance:**
   - Our method addresses key challenges in Meta-RL, such as improving generalization and reducing computational overhead.
   - Empirical results demonstrate faster convergence during meta-training and fewer timesteps required for generalization to out-of-distribution tasks.
   - Comparisons with TrMRL and AMAGO consistently show ECET’s superiority in both efficiency in terms of timesteps and final performance.

---

**Closing Remarks**

We hope that these updates address the concerns raised and clarify the contributions and strengths of our work. The revised manuscript reflects a broader evaluation, a clearer positioning of our method, and a more transparent experimental setup.

We are grateful for the opportunity to further refine our study based on your feedback and look forward to your comments. We hope these revisions meet your expectations and provide a comprehensive basis for evaluating our contributions.

Sincerely,
The Authors

---

### Public Comment · ~Jingyang_You1 · 2026-04-09

Would it be possible for the authors to release the source code for ECET? I cannot seem to replicate the mujoco results. For example, in halfcheetahdir, VariBAD reaches ~1500 using only 4e6 steps, while in the JMLR version of VariBAD, ~1500 can only be obtained with ~4e7 steps, which differs by a factor of 10.

The reason I can come up with is ECET reports the #meta-steps for mujoco environments, instead of the total number of steps. With T=5 and #rollouts=2, the factor 10 is explainable.

The metaworld experiments seem to have reported the total steps.

---

### Meta-Review · Area_Chair_vFZU · 2024-12-22

**Metareview:**

This work proposes an algorithm for learning features for meta-reinforcement learning in latent space. This is achieved by training a two-level hierarchy of transformer networks.  The lower layer represents experienced trajectories while the upper one is trained on all tasks. This method achieves a great good performance in meta-RL benchmarks. It is also demonstrated experimentally that the performance gains are due to the improved representation learning algorithm. Although we had a dissenting reviewer that raised concerns about its novelty and relationship with related work. These were outweighed by the support of other reviewers that highlighted the competitiveness of the empirical results.

**Additional Comments On Reviewer Discussion:**

After the discussion phase there was still discrepancy in the submission's scores. Some reviewers argued for acceptance while others did not. Since the majority opinion is that this work contains meaningful contributions and results, I lean towards recommending acceptance.

---

### Decision · Program_Chairs · 2025-01-22

Accept (Poster)